# Multisite dependency of an E3 ligase controls monoubiquitylation-dependent cell fate decisions

**Achim Werner[1,2†‡], Regina Baur[1,2†], Nia Teerikorpi[2], Deniz U Kaya[2], Michael Rape[1,2]***

[1]Howard Hughes Medical Institute, University of California, Berkeley, Berkeley, United States; [2]Department of Molecular and Cell Biology, University of California, Berkeley, Berkeley, United States

**Abstract** Metazoan development depends on tightly regulated gene expression programs that instruct progenitor cells to adopt specialized fates. Recent work found that posttranslational modifications, such as monoubiquitylation, can determine cell fate also independently of effects on transcription, yet how monoubiquitylation is implemented during development is poorly understood. Here, we have identified a regulatory circuit that controls monoubiquitylation-dependent neural crest specification by the E3 ligase CUL3 and its substrate adaptor KBTBD8. We found that CUL3[KBTBD8] monoubiquitylates its essential targets only after these have been phosphorylated in multiple motifs by CK2, a kinase whose levels gradually increase during embryogenesis. Its dependency on multisite phosphorylation allows CUL3[KBTBD8] to convert the slow rise in embryonic CK2 into decisive recognition of ubiquitylation substrates, which in turn is essential for neural crest specification. We conclude that multisite dependency of an E3 ligase provides a powerful mechanism for switch-like cell fate transitions controlled by monoubiquitylation.
DOI: https://doi.org/10.7554/eLife.35407.001

**\*For correspondence:**
mrape@berkeley.edu

†These authors contributed equally to this work

**Present address:** ‡National Institutes of Dental and Craniofacial Research, NIH, Bethesd, United States

## Introduction

Metazoan development relies on accurate cell fate decisions that give rise to the ~200 cell types of an adult organism. Most differentiation events are switch-like and instruct progenitor cells to decisively adopt a more specialized fate. Epigenetic and transcriptional control have long been known to drive differentiation, and self-reinforcing circuits built around transcription factors or chromatin modifiers help establish the switch-like character of these reactions (*Atlasi and Stunnenberg, 2017*; *Pijnappel et al., 2013*). Recent work suggested that posttranslational modifications, including monoubiquitylation, guide differentiation events also independently of gene expression, yet the underlying mechanisms are incompletely understood.

Monoubiquitylation plays a critical role during craniofacial development (*Jin et al., 2012*; *McGourty et al., 2016*; *Werner et al., 2015*). This process begins with the emergence of neural crest cells, which originate at the border of the neural plate and non-neural ectoderm and migrate to various target tissues in a developing organism (*Betancur et al., 2010*; *Simões-Costa and Bronner, 2015*). Upon reaching their destination, neural crest cells differentiate into various derivatives, including chondrocytes, melanocytes, or glial cells. Cranial neural crest cells emerge at the rostral end of the neural tube, migrate to the branchial arches of the developing head, and differentiate into chondrocytes that lay out a collagen matrix for subsequent bone formation. Accordingly, the misregulation of cranial neural crest specification or differentiation results in pathologies that are characterized by the aberrant formation of craniofacial bone (*Achilleos and Trainor, 2015*).

Generation of neural crest cells relies on the E3 ligase CUL3 and its substrate adaptor KBTBD8 (*Werner et al., 2015*). CUL3$^{KBTBD8}$ monoubiquitylates the RNA-binding proteins TCOF1 and NOLC1, which allows these paralogs to associate with each other and recruit enzymes of ribosome biogenesis and modification. The subsequent production of new and possibly modified ribosomes reshapes the mRNA translation landscape of stem cells in favor of neural crest specification (*Werner et al., 2015*). Underscoring the importance of this regulatory circuit for human biology, mutations in *TCOF1* account for ~90% of cases of the craniofacial disease Treacher Collins Syndrome (*The Treacher Collins Syndrome Colla, 1996*; *Jones et al., 2008*), while additional transcription factor binding sites in the promoter of *KBTBD8* drive melanoma, a cancer of neural crest origin (*Hayward et al., 2017*). At later stages of craniofacial development, CUL3 pairs up with a distinct adaptor, KLHL12, to monoubiquitylate a COPII vesicle coat protein and accelerate collagen secretion (*Jin et al., 2012*; *McGourty et al., 2016*), and mutations in this pathway lead to the craniofacial disorder cranio-lenticulo-sutural dysplasia (*Boyadjiev et al., 2006*). Together, these findings revealed critical roles of monoubiquitylation in cell differentiation and implied that tight regulation of CUL3 is essential for human development.

Despite its importance for neural crest specification, mechanisms that ensure accurate CUL3$^{KBTBD8}$ activation and function are very poorly understood. While CUL3$^{KBTBD8}$ is essential for establishing neural crest cells, it is not required for the maintenance of pluripotent stem cells (*Werner et al., 2015*). This suggested that CUL3$^{KBTBD8}$ engages its targets at specific stages of differentiation, yet how it recognizes its substrates at the right time and place is not known. How monoubiquitylation by CUL3$^{KBTBD8}$ helps TCOF1 and NOLC1 bind each other is also unclear: while monoubiquitylation often recruits effector proteins to a modified target (*Dikic et al., 2009*; *Yau and Rape, 2016*), no ubiquitin-binding domains have been identified in TCOF1, NOLC1, or their known binding partners. Indeed, rather than being organized into structural domains that engage in distinct interactions, TCOF1 and NOLC1 contain large stretches of acidic residues that are predicted to be of low structural complexity (*Lee et al., 2013*). How monoubiquitylation of an intrinsically disordered protein can precipitate a switch-like transition in cellular state is an open question.

Here, we show that CUL3$^{KBTBD8}$-dependent monoubiquitylation and neural crest specification require multisite substrate phosphorylation by CK2, a kinase whose levels gradually increase during development of the nervous system (*Mestres et al., 1994*). The essential CUL3$^{KBTBD8}$-substrates TCOF1 and NOLC1 contain 10 or more motifs that, following their phosphorylation by CK2, can be independently recognized by a conserved surface on KBTBD8. We found that multiple CK2 motifs need to be phosphorylated in the same substrate to mediate both monoubiquitylation by CUL3$^{KBTBD8}$ as well as neural crest specification. Multisite dependency allows cells to convert a gradual increase in kinase input, as seen for embryonic CK2, into decisive activation of signaling output (*Gunawardena, 2005*; *Kapuy et al., 2009*). We therefore propose that multisite dependency of CUL3$^{KBTBD8}$ provides an elegant mechanism for switch-like cell fate decisions controlled by monoubiquitylation.

## Results

### CK2 kinase is required for CUL3$^{KBTBD8}$-dependent neural crest specification

CUL3$^{KBTBD8}$ drives neural crest specification by catalyzing the monoubiquitylation of TCOF1 and NOLC1 (*Werner et al., 2015*), but how it selects its targets at the right time during development is not known. As substrate recognition by cullin-RING ligases often requires posttranslational modifications or co-adaptor proteins (*McGourty et al., 2016*; *Skaar et al., 2013*), we speculated that regulators of CUL3$^{KBTBD8}$ could be identified as shared interactors of NOLC1 and TCOF1. We therefore affinity-purified $^{FLAG}$NOLC1 and $^{FLAG}$TCOF1 from human 293T embryonic kidney cells, a system that had previously allowed us to discover stem cell-related signaling pathways (*Jin et al., 2012*; *McGourty et al., 2016*; *Werner et al., 2015*), and analyzed the immunoprecipitates by CompPASS mass spectrometry (*Huttlin et al., 2015*; *Sowa et al., 2009*). These experiments showed that both NOLC1 and TCOF1 interacted with all subunits of the CK2 kinase (*Figure 1A*), which was consistent with earlier studies that found these proteins to be phosphorylated by CK2 (*Jones et al., 1999*;

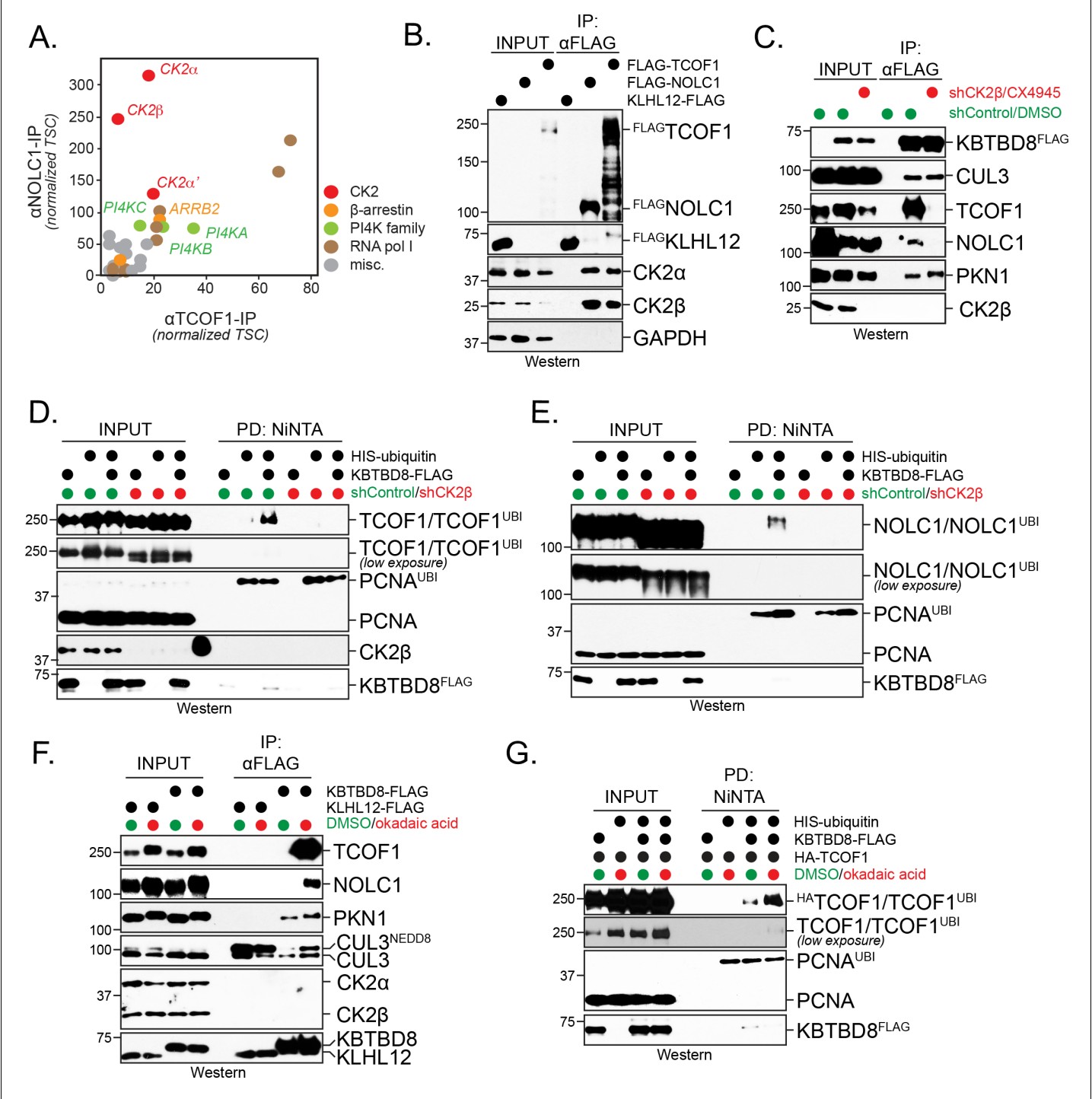

**Figure 1.** CK2 kinase is required for CUL3KBTBD8-substrate binding and ubiquitylation in cells. (**A**) Both NOLC1 and TCOF1 associate with the CK2 kinase. FLAGNOLC1 and FLAGTCOF1 were affinity-purified from 293 T cells and specific binding partners were determined by CompPASS mass spectrometry. Total spectral counts (TSCs) of specific interactors were normalized to 1000 TSCs for each bait protein and plotted against each other. For each CUL3KBTBD8-substrate, three independent affinity-purification and mass spectrometry experiments were performed. The results of each affinity experiments were compared to a database, which included ~100 unrelated affinity-purifications performed with the same antibody and in the same cell line. (**B**) TCOF1 and NOLC1 associate with endogenous CK2 in cells. FLAGNOLC1 and FLAGTCOF1 were affinity-purified from 293 T cells, and co-purifying proteins were identified by western blotting using the indicated antibodies. KLHL12FLAG was used as a negative control. The low molecular weight species seen in TCOF1 samples are proteolytic degradation products caused by partial cleavage of the large and intrinsically disordered TCOF1 during the affinity-purification. (**C**) CK2 is required for substrate recognition by CUL3KBTBD8 in cells. 293 T cells were depleted of CK2β using shRNAs

*Figure 1 continued on next page*

*Figure 1 continued*

and the activity of the remaining CK2 kinase was simultaneously inhibited using the small molecule CX4945. KBTBD8[FLAG], the substrate adaptor of CUL3, was affinity-purified and bound endogenous proteins were detected by western blotting using the indicated antibodies. PKN1 is a specific binding partner of KBTBD8 that is not known to be ubiquitylated by CUL3[KBTBD8] and that is not important for neural crest specification. (D) CK2 is required for monoubiquitylation of TCOF1 in cells. Ubiquitylated proteins were purified under denaturing conditions from 293 T cells expressing [HIS]ubiquitin. Ubiquitylated proteins were detected by western blotting using the indicated antibodies. PCNA is ubiquitylated by a distinct E3 ligase and hence is used as a control for general ubiquitylation efficiency; unmodified PCNA is not detected in NiNTA pulldowns, demonstrating the specificity of this assay. Input: 3% of lysate used for denaturing purification. (E) CK2 is required for monoubiquitylation of NOLC1 in cells. Ubiquitylated NOLC1 was purified both in the absence or presence of CK2 inhibitors, as described for TCOF1 above. Input: 3% of lysate used for denaturing purification. (F) Inhibition of PP1 and PP2A phosphatases, which counteract CK2 in cells, promotes substrate binding by CUL3[KBTBD8]. KBTBD8[FLAG] or the negative control KLHL12[FLAG] were affinity-purified from cells treated with the PP1/PP2A-inhibitor okadaic acid. Co-purifying proteins, including endogenous TCOF1, NOLC1, and the control KBTBD8-interactor PKN1, were detected by western blotting. This experiment was performed at smaller scale than *Figure 1C*, explaining the lack of TCOF1 binding to KBTBD8 under conditions without phosphatase inhibitors. (G) Inhibition of PP1/PP2A phosphatases increases TCOF1 ubiquitylation in cells. Ubiquitylated [HA]TCOF1 was purified under denaturing conditions from 293 T cells expressing [HIS]ubiquitin and treated with or without okadaic acid. Ubiquitylated proteins were detected by Western blotting. Input: 3% of lysate used for denaturing purification.

DOI: https://doi.org/10.7554/eLife.35407.002

The following figure supplement is available for figure 1:

**Figure supplement 1.** Expression analysis of CK2 and PP1/PP2A in differentiating hESCs.

DOI: https://doi.org/10.7554/eLife.35407.003

*Meier and Blobel, 1992*; *Wise et al., 1997*). We confirmed the robust interaction of NOLC1 and TCOF1 with CK2α and CK2β by affinity-purification and western blotting (*Figure 1B*).

Expression of CK2 gradually increases during embryogenesis at times of neural crest development (*Mestres et al., 1994*), and CK2 activity had been detected in the branchial arches that are important for craniofacial bone development (*Jones et al., 1999*). Together with our binding studies, these observations suggested that phosphorylation by CK2 might play a role in target recognition and monoubiquitylation by CUL3[KBTBD8]. To test this hypothesis, we purified KBTBD8, the subunit of CUL3[KBTBD8] that directly binds substrates, from cells that were transfected with shRNAs against CK2β, that is the subunit shared by both CK2a/β and CK2α'/β complexes, and treated with a small molecule CK2 inhibitor (CX4945). Such effective inhibition of CK2 prevented NOLC1 or TCOF1 from binding to CUL3[KBTBD8], whereas PKN1, a protein that associates with a different surface of KBTBD8 and is not important for neural crest specification (*Werner et al., 2015*), was still recognized (*Figure 1C*). Inhibition of CK2 also blocked CUL3[KBTBD8]-dependent monoubiquitylation of TCOF1 and NOLC1, while the modification of an unrelated protein, PCNA, was not impaired (*Figure 1D,E*). In line with these results, inactivation of PP1 and PP2A phosphatases, which oppose CK2 in cells (*Popescu et al., 2009*), strongly promoted substrate binding and monoubiquitylation by CUL3[KBTBD8] (*Figure 1F,G*). Interestingly, the phosphatase subunit PP1A decreases during neural crest specification in vitro at times when TCOF1 and NOLC1 engage CUL3[KBTBD8] (*Figure 1—figure supplement 1A,B*).

Monoubiquitylation by CUL3[KBTBD8] allows NOLC1 and TCOF1 to bind each other and recruit enzymes required for ribosome biogenesis and neural crest specification (*Werner et al., 2015*). To determine whether CK2 is required for CUL3[KBTBD8]-dependent signaling, we therefore asked whether CK2 inhibition impaired formation of TCOF1-NOLC1 complexes. As seen before (*Werner et al., 2015*), we found that the expression of KBTBD8 caused TCOF1 and NOLC1 to associate with each other (*Figure 2A*). Loss of CK2 prevented the CUL3[KBTBD8]-dependent assembly of the TCOF1-NOLC1 platform to the same extent as inactivation of KBTBD8 through mutation of its CUL3-binding motif (KBTBD8[Y74A]) (*Figure 2A*). Accordingly, depletion of CK2β or small molecule inhibition of CK2 kinase activity interfered with neural crest specification (*Figure 2B–D*). We had previously shown that hESCs that fail to produce neural crest cells induce a compensatory pathway characterized by increased expression of forebrain markers, FOXG1 and SIX3 (*Werner et al., 2015*), and we observed the same response if hESCs were subjected to differentiation in the absence of CK2 (*Figure 2C*). By contrast, low concentrations of the PP1 and PP2A inhibitor okadaic acid increased the efficiency of neural crest specification (*Figure 2—figure supplement 1*). We conclude that CK2 is required for the recognition and monoubiquitylation of TCOF1 and NOLC1 by CUL3[KBTBD8], formation of a TCOF1-NOLC1 ribosome biogenesis platform, and neural crest specification.

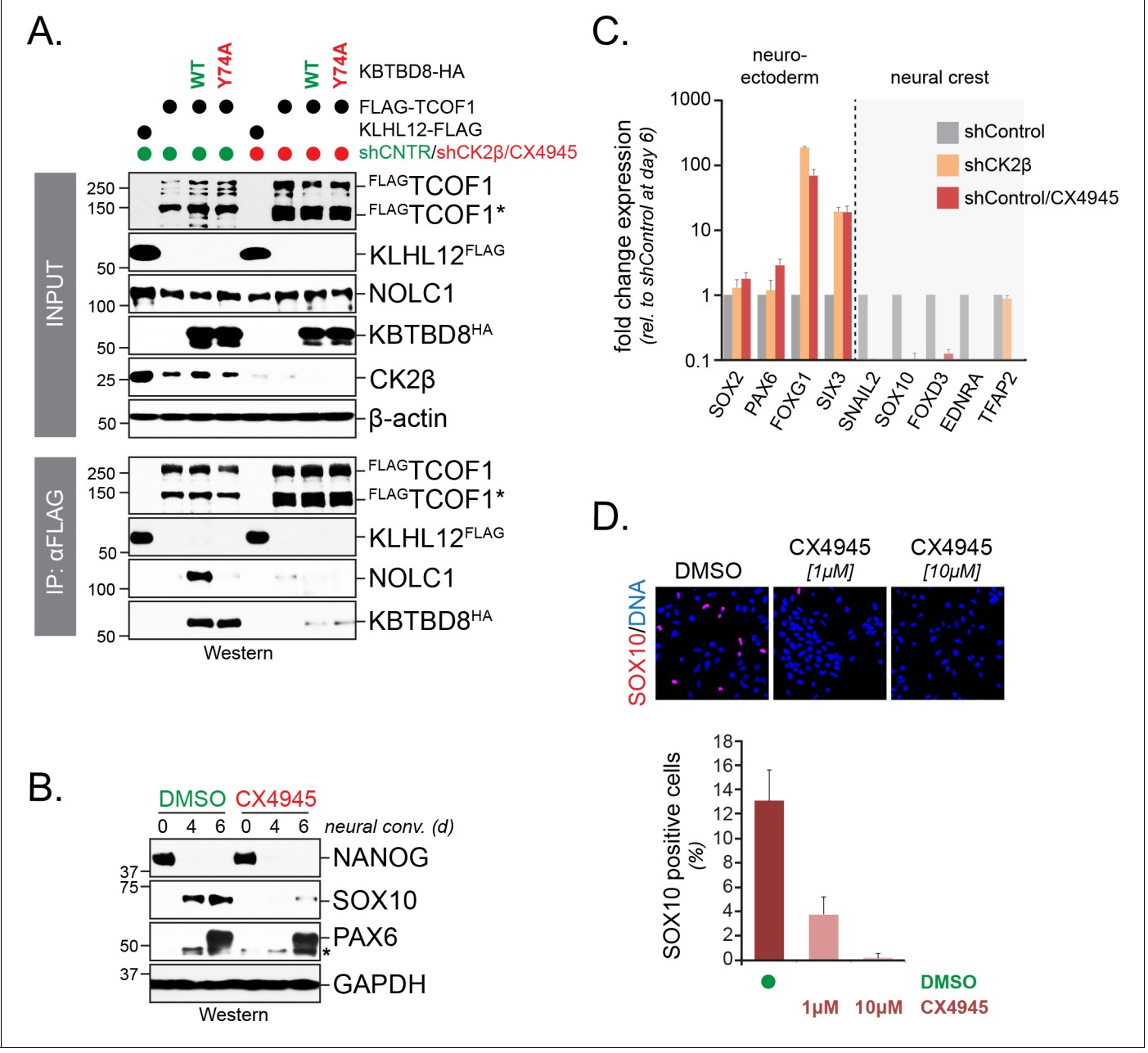

**Figure 2.** CK2 is required for neural crest specification. (**A**) CK2 is required for CUL3[KBTBD8]- and ubiquitylation-dependent formation of TCOF1-NOLC1 complexes. [FLAG]TCOF1 was affinity-purified from control- or CK2 activity-depleted cells (shCK2β/CX4945) that expressed WT-KBTBD8 or mutant KBTBD8[Y74A], which is unable to bind CUL3 and thus cannot catalyze monoubiquitylation. Bound proteins, including endogenous NOLC1, were detected by western blotting. KLHL12[FLAG] was used as a control protein. (**B**) CK2 activity is required for neural crest formation from hESCs. H1 hESCs were subjected to neural conversion in the absence or presence of 1 μM CK2 inhibitor CX4945. Expression of markers for central nervous system precursor (PAX6) or the neural crest (SOX10) was analyzed by Western blotting. Differentiation was monitored by following the stem cell-specific transcription factor NANOG, using specific antibodies. (**C**) CK2 controls a switch between CNS and neural crest precursors. Neural conversion of H1 hESCs was induced under control conditions (shControl), in the presence of an shRNA to deplete endogenous CK2β (shCK2β), or in the presence of 1 μM CK2 inhibitor CX4945 (shControl/CX4549). Differentiation was followed by qRT-PCR analysis of markers of the CNS (SOX2, PAX6, FOXG1, SIX3) or the neural crest (SNAIL2, SOX10, FOXD3, EDNRA, TFAP2). Fold change expression is relative to shControl at day 6 of differentiation. Error bars denote s.e.m. of three technical replicates. (**D**) CK2 is required for neural crest specification, as detected at the single cell level. H1 hESCs were subjected to neural conversion in the absence of presence of indicated concentrations of CK2 inhibitor CX4945 for 7d. Formation of neural crest cells was assessed by indirect immunofluorescence microscopy using antibodies specific to SOX10. Representative images are shown. SOX10 positive cells were quantified from three independent experiments and expressed as percentage of total cells imaged. Error bars denote standard deviation.
DOI: https://doi.org/10.7554/eLife.35407.004

*Figure 2 continued on next page*

*Figure 2 continued*

The following figure supplement is available for figure 2:

**Figure supplement 1.** Phosphatase inhibition increases the efficiency of neural crest specification.

DOI: https://doi.org/10.7554/eLife.35407.005

## A conserved loop in KBTBD8 is required for CK2 substrate recognition

By phosphorylating residues embedded in acidic sequences, CK2 generates highly negatively charged proteins (*Bian et al., 2013*; *Meggio and Pinna, 2003*). As CK2 appeared to drive target recognition by CUL3$^{KBTBD8}$, we expected that a complementary positively charged motif in KBTBD8 might mediate substrate binding. By investigating sequence conservation among 54 vertebrate KBTBD8 homologs, we found a highly conserved positively charged loop that was in proximity of Trp579 (*Figure 3A*; *Figure 3—figure supplement 1*), a residue that is known to be required for substrate binding by CUL3$^{KBTBD8}$ (*Werner et al., 2015*). Replacing five His and Lys residues within this loop with Ala (KBTBD8$^{5HK}$) prevented binding of KBTBD8 to NOLC1 and TCOF1, whereas the interaction of KBTBD8 with CUL3 or PKN1 was not affected by this charge removal (*Figure 3B*). Similar observations were made by affinity-purification coupled to mass spectrometry, where mutation of its positively charged loop blocked KBTBD8-binding to TCOF1 or NOLC1 (*Figure 3C*).

In line with these results, KBTBD8$^{5HK}$ was an inactive substrate adaptor that was unable to promote NOLC1 or TCOF1 ubiquitylation in cells (*Figure 3D,E*) and failed to induce formation of the NOLC1-TCOF1 complex (*Figure 3F*). Moreover, if KBTBD8$^{5HK}$ was the only KBTBD8 variant expressed in hESCs, cells did not differentiate into neural crest cells and instead showed the compensatory induction of the forebrain markers FOXG1 and SIX3 upon neural conversion (*Figure 3G, H*). Thus, a positively charged loop in KBTBD8 is required for CUL3$^{KBTBD8}$ substrate monoubiquitylation and neural crest specification. This strongly suggested that CK2 phosphorylates CUL3$^{KBTBD8}$-substrates to mark them for monoubiquitylation.

## CK2 is required for NOLC1 and TCOF1 recognition by CUL3$^{KBTBD8}$

To test the above hypothesis, we purified TCOF1 and NOLC1 fragments that contained two CK2 consensus motifs with several phosphorylation sites each. As evidenced by a CK2- and ATP-dependent increase in molecular weight, these proteins were efficiently phosphorylated by the CK2α/β holoenzyme in vitro (*Figure 4A*). Importantly, when we then incubated purified KBTBD8 with either unmodified or phosphorylated TCOF1 or NOLC1, we found that KBTBD8 selectively bound the phosphorylated, but not unmodified, targets (*Figure 4B,C*). KBTBD8 did not recognize phosphorylated substrates if its Trp579 or the positively charged 5HK-patch were mutated (*Figure 4B,C*), and KBTBD8 also did not bind a TCOF1 fragment, whose CK2 motif was destroyed by introduction of positively charged Arg residues or mutation of seven phosphoacceptor Ser residues (*Figure 4D*). By contrast, Phe550 of KBTBD8, which is required for recognition of PKN1 (*Werner et al., 2015*), was not needed for the interaction between phosphorylated TCOF1 or NOLC1 and KBTBD8 (*Figure 4B, C*).

Titration experiments relying on fluorescence polarization revealed that KBTBD8 bound a single phosphorylated CK2 motif with a $K_D$ of ~1.4 μM, while we did not detect any interaction with an unphosphorylated TCOF1 peptide (*Figure 4E*). A monomeric variant of KBTBD8, in which residues at the BTB interface were mutated to prevent adaptor dimerization (*Zhuang et al., 2009*), bound the phosphorylated CK2 motif with a similar affinity as the wildtype dimeric KBTBD8 (*Figure 4E*), whereas mutation of its positively charged surface (KBTBD8$^{5HK}$) or competition with a negatively charged molecule known to interfere with CK2 substrate recognition, IP6, prevented substrate binding by KBTBD8 (*Figure 4F,G*). The latter finding underscores the importance of electrostatic interactions for substrate recognition by CUL3$^{KBTBD8}$.

Having observed the direct binding of phosphorylated TCOF1 to KBTBD8, we were able to reconstitute CUL3$^{KBTBD8}$-dependent substrate monoubiquitylation in a purified setting. We first allowed KBTBD8 to bind a phosphorylated TCOF1 fragment with two CK2-motifs, a step that was required for specific substrate modification. We then added neddylated CUL3-RBX1, E1, E2, and ubiquitin and found that the phosphorylated TCOF1 was efficiently monoubiquitylated (*Figure 4H*). No ubiquitin transfer was detected in the absence of ATP or KBTBD8 (*Figure 4H*); moreover, we

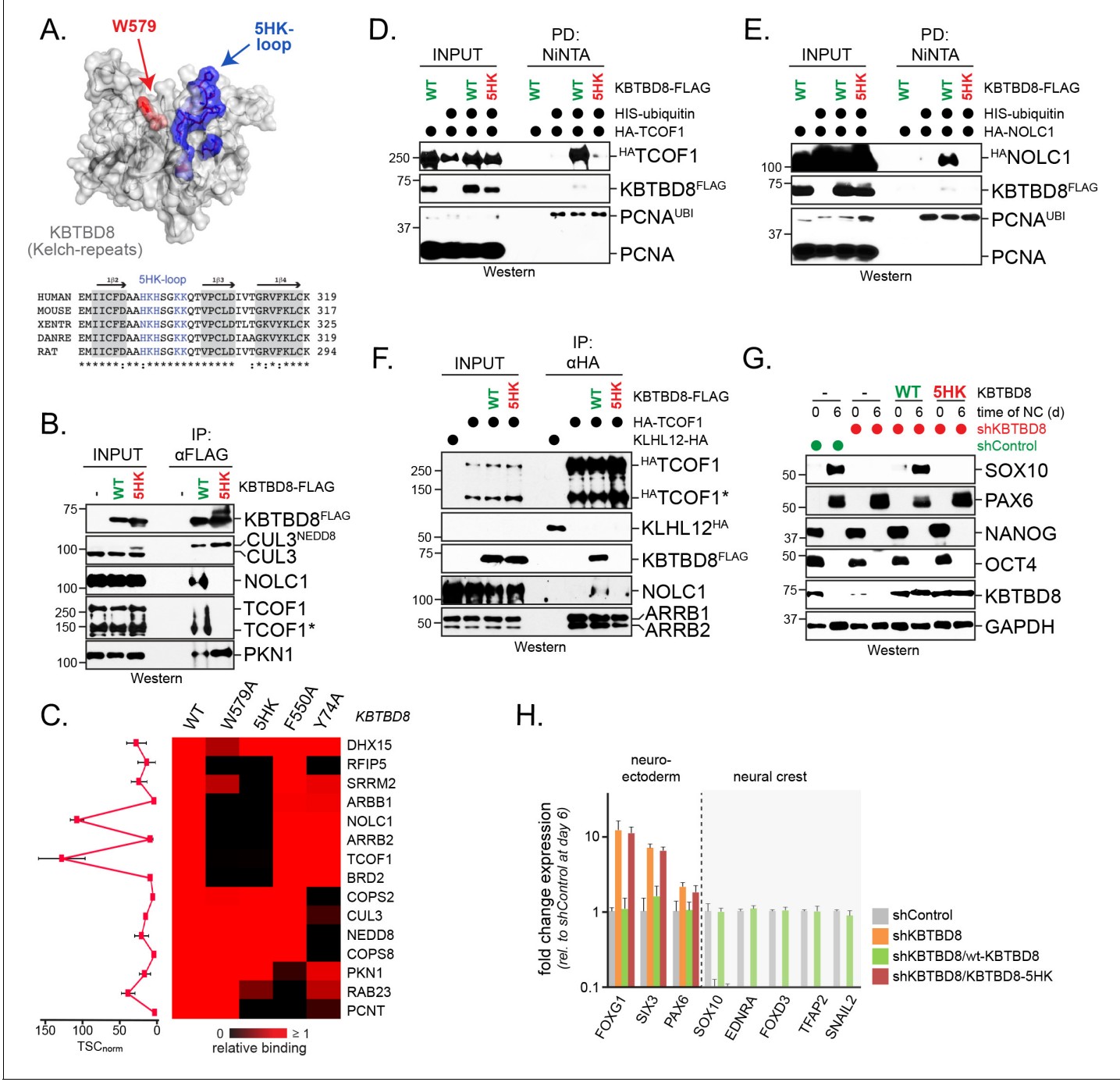

**Figure 3.** A conserved loop in KBTBD8 is required for CK2-substrate recognition. (A) KBTBD8 contains a loop with five His- or Lys-residues that is close to W579, a KBTBD8-residue known to be important for substrate recognition. Depicted is a structural model of the Kelch-domain of KBTBD8; W579 is shown in red, whereas the 5HK-loop is shown in blue. (B) The 5HK-loop in KBTBD8 is required for substrate recognition. KBTBD8[FLAG] or KBTBD8[5HK-FLAG] (i.e. a KBTBD8-mutant in which all five His- or Lys-residues in the conserved loop were exchanged to Ala) were affinity-purified from cells. Co-purifying TCOF1, NOLC1, or PKN1 were detected by western blotting. (C) The 5HK-loop of KBTBD8 is essential for substrate recognition. The indicated KBTBD8 variants (wt; W579A; 5HK-Ala; F550A; Y74A) were affinity-purified from 293 T cells and analyzed for binding partners by CompPASS mass spectrometry. The heatmap depicts the relative binding of interactors identified for wild-type KBTBD8 in the respective KBTBD8 variant (black = no interaction, red = equal or more interaction). A quantification of normalized total spectral counts (TSCs) of three biological replicates is shown for the wild-type KBTBD8 affinity-purification is shown on the left. (D) The 5HK-loop of KBTBD8 is required for TCOF1 ubiquitylation in cells. Ubiquitylated proteins were purified under denaturing conditions from 293 T cells expressing [HIS]ubiquitin. Modified proteins were detected by western blotting, using the indicated antibodies. (E) The 5HK-loop of KBTBD8 is required for NOLC1 ubiquitylation in cells. Ubiquitylated proteins were purified from 293

*Figure 3 continued on next page*

*Figure 3 continued*

T cells under denaturing conditions and analyzed as described above. (F) The 5HK loop of KBTBD8 is required for TCOF1-NOLC1 complex formation. Cells expressing $^{HA}$TCOF1 and either KBTBD8$^{FLAG}$, KBTBD8$^{5HK-FLAG}$, or the control protein KLHL12$^{FLAG}$ were subjected to anti-HA affinity-purification and analyzed for co-purifying proteins by western blotting. ARBB1 and ARBB2 binding to TCOF1 shows that the efficiency of affinity-purification was identical under all conditions. (G) The 5HK loop in KBTBD8 is required for neural crest specification. H1 hESCs were depleted of endogenous KBTBD8 using stably expressed shRNAs. Where indicated, hESCs were reconstituted with shRNA-resistant wild-type (WT) or mutant (5HK) KBTBD8. Differentiation was followed be western blotting using markers of the neural crest (SOX10), CNS precursors (PAX6), or pluripotency (NANOG; OCT4). (H) The 5HK loop of KBTBD8 is required for neural crest specification. H1 hESCs were depleted of KBTBD8 and reconstituted with shRNA-resistant wild-type KBTBD8 or mutant KBTBD8$^{5HK}$. Expression of markers of CNS/forebrain (FOXG1, SIX3, PAX6) or the neural crest (SOX10, EDNRA, FOXD3, TFAP2, SNAIL2) was followed by qRT-PCR. Fold change expression is relative to shControl at day 6 of differentiation. Error bars denote s.e.m. of three technical replicates.

DOI: https://doi.org/10.7554/eLife.35407.006

The following figure supplement is available for figure 3:

**Figure supplement 1.** Conservation analysis of KBTBD8.

DOI: https://doi.org/10.7554/eLife.35407.007

observed a strong inhibition of ubiquitylation in the presence of KBTBD8 variants that were less efficient in binding TCOF1, including KBTBD8$^{5HK}$ or KBTBD8$^{W579A}$ (*Figure 4I*). We conclude that phosphorylation by CK2 marks TCOF1 and NOLC1 for recognition and monoubiquitylation by CUL3$^{KBTBD8}$.

## TCOF1 and NOLC1 contain multiple recognition motifs for CUL3$^{KBTBD8}$

Interestingly, while KBTBD8$^{W579A}$ did not bind TCOF1 fragments with two or more CK2 motifs (*Figure 4B*), it associated with a peptide containing a single phosphorylated CK2 motif (*Figure 4F*). This implied that Trp579 stabilizes a conformation of KBTBD8 that is required for this E3 to engage more than one negatively charged CK2 motif at the same time. Consistent with this, we noticed that TCOF1 and NOLC1 each have more than 10 CK2 motifs that are embedded in the central acidic domain of low structural complexity (*Figure 5A*). To assess whether these motifs provide CUL3$^{KBTBD8}$ recognition elements, we purified 24 distinct CK2 repeats from TCOF1 and NOLC1 and tested whether they could bind KBTBD8 in vitro. We found that KBTBD8 associated in a CK2-dependent manner with 10 CK2 motifs in TCOF1 and nine counterparts in NOLC1 (*Figure 5B*). These results were supported by cellular studies: TCOF1 was monoubiquitylated on distinct Lys residues that were in close proximity to nine different phosphorylated CK2 motifs (*Figure 5C*) (*Bian et al., 2013*). TCOF1 and NOLC1 therefore do not contain a single, but many recognition motifs for CUL3$^{KBTBD8}$.

## Multisite dependency of CUL3$^{KBTBD8}$ substrate recognition

Why do TCOF1 and NOLC1 have so many recognition motifs for their E3? Multiple E3 binding sites might ensure robust substrate ubiquitylation, as backup sites could buffer mutation or defective phosphorylation of a single motif. If this were to be the case, increasing the number of CK2 motifs should not improve substrate affinity under conditions of saturating kinase activity. Alternatively, studies on the cyclin-dependent kinase inhibitor SIC1 found that multiple E3-binding sites could produce allovancy (*Csizmok et al., 2017*; *Nash et al., 2001*; *Tang et al., 2007*): following dissociation of one SIC1-degron, neighboring E3 recognition motifs already in proximity to the enzyme rapidly rebind and thereby retain the interaction between the substrate and its E3. It would also be possible that multiple CK2 motifs access different positively charged patches on KBTBD8 in a multivalent manner. If the latter two mechanisms were at play for CUL3$^{KBTBD8}$, adding more CK2 sites would be expected to increase the affinity of substrates for this E3. Supporting these models, we found in a competition assay that the more CK2 sites were included in TCOF1, the better it bound KBTBD8 (*Figure 5D,E*; *Figure 5—figure supplement 1*). We noted a particularly prominent increase in substrate affinity if TCOF1 fragments contained six or more CK2 sites (*Figure 5D,E*).

We confirmed these results by monitoring the ability of immobilized KBTBD8 to retain TCOF1 fragments with increasing numbers of CK2 motifs. When assayed at high substrate concentration, TCOF1 fragments with 2, 3, 4, 6, and 8 CK2 motifs were all able to bind KBTBD8 in a CK2-dependent manner (*Figure 5F*). However, if all peptides were incubated with KBTBD8 at the same time,

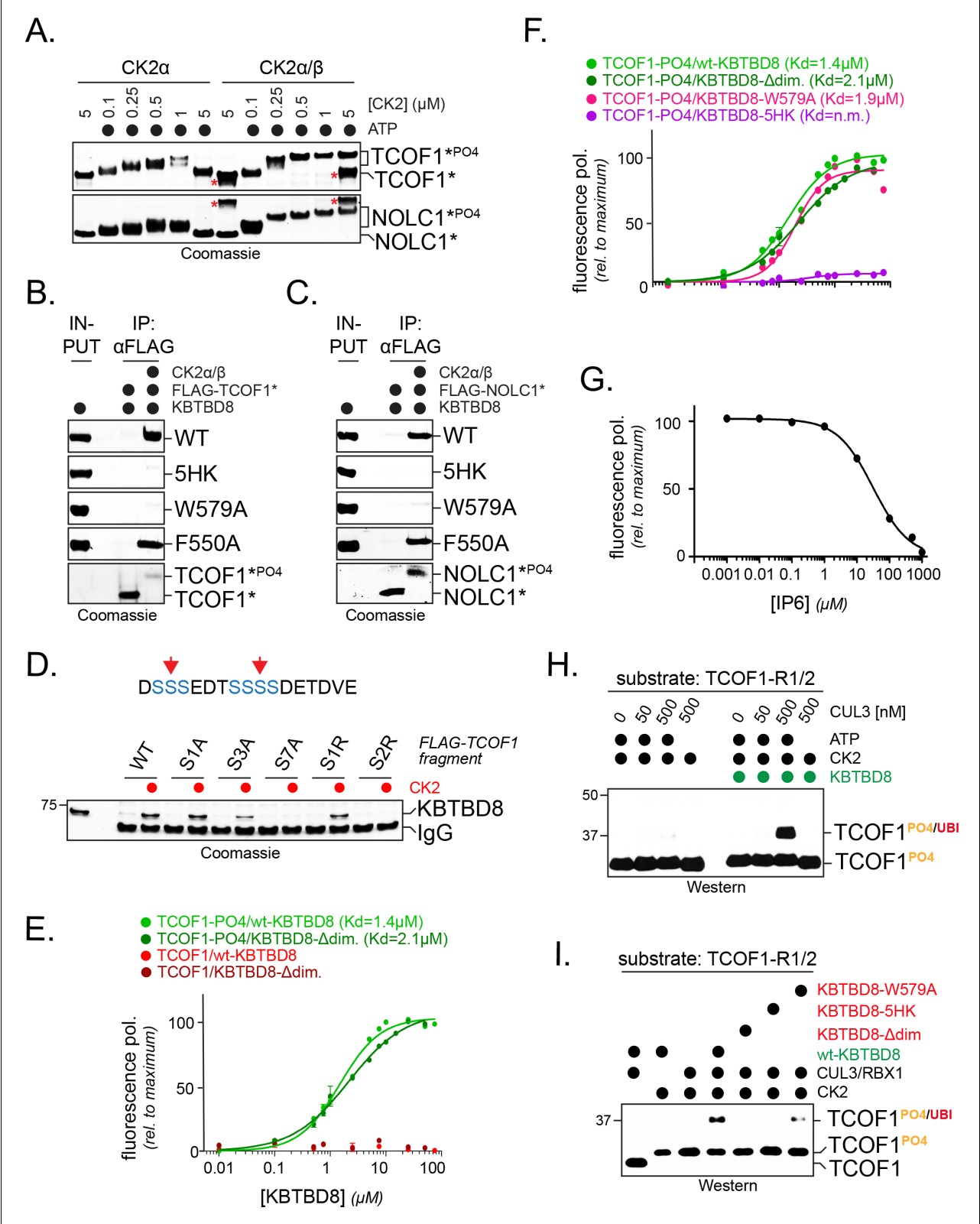

**Figure 4.** CK2 marks CUL3[KBTBD8]-substrates for monoubiquitylation in vitro. (**A**) TCOF1 and NOLC1 are phosphorylated by recombinant CK2 in vitro. Recombinant Flag-tagged TCOF1[190-364] or NOLC1[76-187] fragments with two CK2 motifs (Flag-TCOF1*, Flag-NOLC1*) were incubated with either the catalytic subunit CK2α or complexes between CK2α and the regulatory subunit CK2β. ATP was added as indicated, and proteins were detected by Coomassie staining. The molecular weight shift observed in the presence of both CK2 and ATP is indicative of phosphorylation. The red asterisk marks

*Figure 4 continued on next page*

Figure 4 continued

CK2β, as detected at its highest concentration. In the absence of the stabilizing CK2β-subunit, high concentrations of CK2α likely lead to aggregation and reduced TCOF1 phosphorylation. (B) KBTBD8 selectively binds phosphorylated TCOF1. The binding of recombinant Flag-TCOF1* (containing two CK2 motifs) to KBTBD8 was analyzed in the absence or presence of CK2α/β. TCOF1 was immobilized on beads and bound KBTBD8 was detected by Coomassie staining. The KBTBD8 variants tested (WT, 5HK, W579A, F550A) are indicated on the right. (C) KBTBD8 selectively binds phosphorylated NOLC1. The binding of recombinant Flag-NOLC1* (containing two CK2 motifs) to KBTBD8 was analyzed as described above for TCOF1. (D) Mutation of a CK2 motif obliterates its recognition by KBTBD8. The indicated CK2 motif of TCOF1[197-213] was altered by introduction of one or two Arg residues (sites indicated by red arrows in scheme above) or by mutation of one (S1A), three (S3A) or all seven (S7A) phosphoacceptor Ser residues. Binding to immobilized KBTBD8 was analyzed either in the absence or presence of CK2, using Coomassie staining. (E) KBTBD8 binds a CK2 motif in TCOF1 in a phosphorylation-dependent manner. A single N-terminally TAMRA-labeled CK2 motif of TCOF1[194-216] was either incubated with CK2 ('-PO4') or kept unmodified and incubated with increasing concentrations of recombinant KBTBD8. Binding of KBTBD8 to the CK2 motif was measured as change in fluorescence polarization, and the dissociation constant of the binding reaction was determined by fitting to a two-state binding curve. In addition to wild-type KBTBD8, the ability of a monomeric variant of KBTBD8 (KBTBD8-Δdim) was monitored (green curves). In the absence of CK2, no measurable binding was detected (red curves). (F) Mutation of the 5HK loop, but not of W579, obliterates binding of KBTBD8 to a single CK2 motif. Binding of distinct KBTBD8 variants (WT, Δdim, W579A, 5HK) to a phosphorylated CK2 motif of TCOF1 was measured by fluorescence polarization. (G) Substrate binding by KBTBD8 is likely electrostatic in nature. Binding of a single phosphorylated CK2 motif from TCOF1 to recombinant KBTBD8 was measured by fluorescence polarization in the presence of increasing concentrations of the highly negatively charged molecule inositolehexakisphosphate (IP6). (H) TCOF1 is monoubiquitylated in vitro. A TCOF1 fragment containing two CK2 motifs ('R1/2') was first phosphorylated by CK2α/β and then immobilized on FLAG-agarose followed by incubation with KBTBD8; immobilization interfered with non-specific ubiquitylation of the substrate peptide. Next, recombinant neddylated CUL3/Rbx1, E1, UBE2D3, ubiquitin, and ATP were added as indicated, and ubiquitylation of TCOF1 was followed by Western blotting. (I) Monoubiquitylation of TCOF1-R1/2 by CUL3[KBTBD8] was analyzed as described above, using the indicated KBTBD8 mutants.

DOI: https://doi.org/10.7554/eLife.35407.008

only TCOF1 fragments with six or more CK2 motifs were retained by the E3 (*Figure 5F*). When tested at substrate concentrations below the K$_D$ of a single CK2 motif, KBTBD8 only bound fragments with multiple CK2 motifs even in the absence of competition (*Figure 5G*). Under these conditions, KBTBD8 required dimerization for target recognition (*Figure 5G*), which was also seen with KBTBD8 immunoprecipitated from cells (*Figure 5H*), as well as in in vitro ubiquitylation assays (*Figure 4I*). In contrast to the effects on affinity, increasing the number of E3-binding sites did not alter the nature of ubiquitylation, and a TCOF1-fragment with eight CK2 motifs was still only monoubiquitylated by CUL3[KBTBD8] (*Figure 5I*). Thus, CUL3[KBTBD8] requires multiple CK2 motifs for optimal substrate recognition, a feature that is referred to as multisite dependency.

Multisite dependency allows cells to translate small increases in the levels or activity of a kinase into large changes in signaling output (*Csizmok et al., 2016*; *Gunawardena, 2005*), which can introduce ultrasensitivity needed for switch-like differentiation (*Serber and Ferrell, 2007*). To test whether its mode of substrate recognition enables CUL3[KBTBD8] to convert the gradual increase in CK2 levels during embryogenesis into decisive recognition of NOLC1 or TCOF1, we titrated CK2α/β into reactions that monitored the binding of a TCOF1 fragment with eight CK2 motifs to KBTBD8; to mimic the cellular situation, where many proteins compete for access to CK2 (*Bian et al., 2013*), we also included an excess competitor with two CK2 motifs. Notably, we found that in the presence of the competitor, TCOF1 was only able to bind KBTBD8 at high CK2 levels, where the affinity gain provided by eight phosphorylated CK2 motifs overcame the concentration advantage of the competitor (*Figure 5J*). Accordingly, the Hill coefficient of the TCOF1-KBTBD8 binding reaction increased from ~1 in the absence of a competitor to ~3.5 in its presence, which is indicative of ultrasensitive behavior. Based on these observations, we infer that multisite dependency of CUL3[KBTBD8] provides a mechanism for cells to translate a slow increase in CK2 levels into switch-like E3 substrate recognition.

## Multisite dependency by CUL3[KBTBD8] controls substrate recognition in cells

We finally wished to determine whether multisite dependency governed substrate regulation by CUL3[KBTBD8] in cells. Different from in vitro conditions, we could not address this question by expressing TCOF1 or NOLC1 fragments with an increasing number of CK2 motifs, as truncated polypeptides would miss interaction surfaces for proteins involved in ribosome biogenesis and neural crest specification (*Lin and Yeh, 2009*; *Prieto and McStay, 2007*). Instead, we decided to inactivate distinct CK2 motifs in full-length NOLC1, which could be introduced into cells more reproducibly

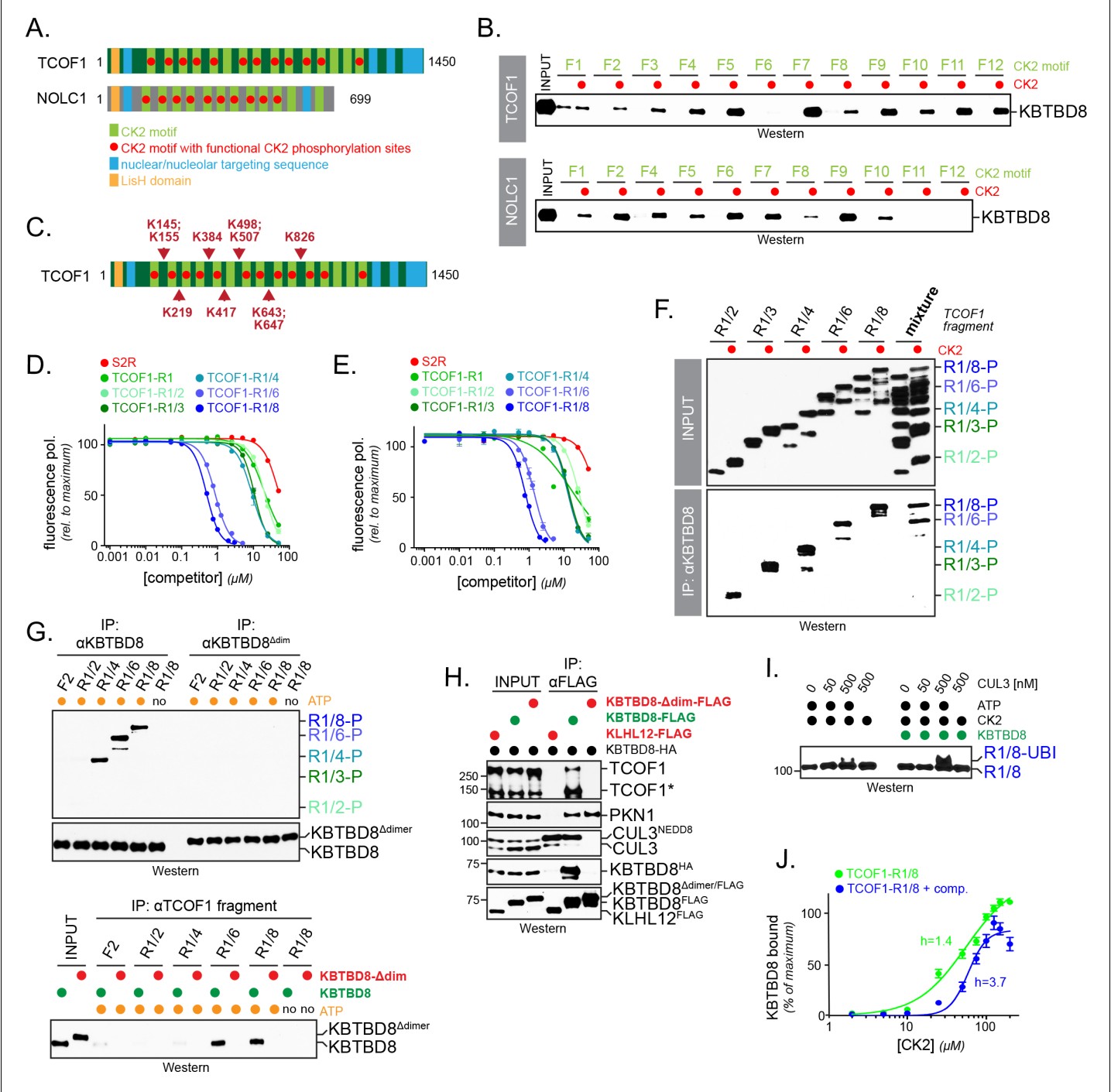

**Figure 5.** Multisite dependency of substrate recognition by CUL3[KBTBD8] in vitro. (**A**) TCOF1 and NOLC1 contain multiple CK2 motifs (light green). Functional motifs are marked with a red dot (non-functional motifs typically contain Arg substitutions that are incompatible with recognition by CK2). The LisH domain is a protein interaction module found in many proteins, but often with unknown function. (**B**) Multiple CK2 motifs are recognized by KBTBD8. TCOF1 or NOLC1 fragments containing single CK2 motifs (number according to position in sequence) were incubated with buffer or with CK2/ATP, immobilized on beads, and incubated with recombinant KBTBD8. Bound KBTBD8 was visualized by western blotting using a specific antibody. (**C**) TCOF1 is ubiquitylated in vivo on multiple Lys residues that are all in close proximity to CK2 motifs. Ubiquitylated TCOF1 was purified under denaturing conditions from 293 T cells and ubiquitylated Lys residues were determined by mass spectrometry. The figure depicts the location of ubiquitylated lysine residues identified in individual peptides. (**D**) The more CK2 motifs are in a substrate, the better it binds to KBTBD8. A single phosphorylated and fluorescently labeled CK2 motif of TCOF1 was bound to KBTBD8. Unlabeled TCOF1 competitor fragments containing either two (R1/2), three (R1/3), four (R1/4), six (R1/6) or eight (R1/8) CK2 motifs were phosphorylated and titrated into the binding reaction, and dissociation of the

*Figure 5 continued on next page*

*Figure 5 continued*

reporter peptide was monitored by loss of fluorescence polarization. (**E**) The increase in affinity afforded by multiple CK2 motifs also applies to monomeric KBTBD8. A monomeric variant of KBTBD8 was produced by mutation of conserved residues at the BTB dimer interface, as described for KLHL12 (*McGourty et al., 2016*). Binding of a single phosphorylated CK2 motif of TCOF1 to monomeric KBTBD8 was then monitored in the presence of competitor peptides by fluorescence polarization, as described above. (**F**) Multiple CK2 motifs increase substrate affinity for KBTBD8. TCOF1 fragments containing either two (R1/2), three (R1/3), four (R1/4), six (R1/6) or eight (R1/8) CK2 motifs were incubated with immobilized KBTBD8 either in the presence or absence of CK2. In the last two lanes, all TCOF1 fragments were mixed and incubated at the same time with KBTBD8. Binding was monitored by western botting (top lanes: input; bottom lanes: KBTBD8-immunoprecipitation). (**G**) Multiple CK2 motifs increase substrate affinity towards KBTBD8 at low substrate concentrations. *Upper panel:* 450 nM of TCOF1 fragments with a single (F2) or multiple (R1/2, R1/4, R1/8) CK2 motifs were incubated with either wild-type KBTBD8 or monomeric KBTBD8$^{\Delta\text{dim}}$, and KBTBD8 was then affinity-purified using a specific antibody. *Lower panel:* Same binding reaction as above, but TCOF1 was immobilized on beads. Complex formation was analyzed by western blotting. (**H**) Dimerization of KBTBD8 is required for TCOF1, but not PKN1, binding in cells. Cells expressing indicated KBTBD8 variants were subjected to KBTBD8$^{\text{FLAG}}$ affinity-purification and analyzed for dimerization (KBTBD8$^{\text{HA}}$), CUL3-binding, or substrate recognition (TCOF1, PKN1) using Western blotting with specific antibodies. (**I**) CUL3$^{\text{KBTBD8}}$-substrates with multiple CK2 motifs are monoubiquitylated. Ubiquitylation of TCOF1-R1/8, a fragment containing eight CK2 motifs, was analyzed in vitro using purified neddylated CUL3-RBX1, KBTBD8, E1, UBE2D3, ubiquitin and ATP, as described above. (**J**) Ultrasensitivity of the KBTBD8-TCOF1 interaction. Binding of TCOF1-R1/8 to KBTBD8 was analyzed in the presence of increasing CK2 concentrations, as described in the *Figure 5F*. Western blots of three independent experiments were quantified by Image J, and error bars denote standard deviation. Green curve: binding reaction in the absence of competitor; blue curve: binding of TCOF1-R1/8 in the presence of a ten-fold excess of TCOF1-R1/2, which mimics endogenous CK2 substrates that typically have one or two CK2 motifs.

DOI: https://doi.org/10.7554/eLife.35407.009

The following figure supplement is available for figure 5:

**Figure supplement 1.** N- and C-terminal TCOF1 fragments both require multiple CK2 motifs for recognition.
DOI: https://doi.org/10.7554/eLife.35407.010

than the much larger TCOF1. We then tested how many functional CK2 motifs were required for NOLC1 recognition by CUL3$^{\text{KBTBD8}}$.

Building off of our observation that introduction of Arg residues destroyed CK2 motifs (*Figure 4D*), we first generated a NOLC1 variant in which all but one CK2 consensus sites were inactivated (NOLC1$^{\text{CK1}}$). Contrary to the wildtype protein, NOLC1$^{\text{CK1}}$ failed to bind endogenous KBTBD8 either in 293 T cells or in differentiating hESCs (*Figure 6A,B*). Thus, although a single CK2 motif can mediate binding of NOLC1 to KBTBD8 in vitro (*Figure 5B*), it is not sufficient to mediate its CUL3$^{\text{KBTBD8}}$-recognition in vivo. We then reintroduced CK2 motifs into NOLC1$^{\text{CK1}}$ to generate NOLC1$^{\text{CK3}}$ (i.e. NOLC1 with three intact CK2 consensus sites), NOLC1$^{\text{CK5}}$, NOLC1$^{\text{CK7}}$, and NOLC1$^{\text{CK9}}$ (*Figure 6C*). Upon analyzing the behavior of these variants toward endogenous or over-expressed KBTBD8, we found that multiple CK2 repeats were required for an interaction between NOLC1 and KBTBD8, and wild-type NOLC1 with 10 CK2 motifs showed the strongest binding to KBTBD8 (*Figure 6C,E*). In line with these results, we found that CUL3$^{\text{KBTBD8}}$-dependent monoubiquitylation of NOLC1 and its ubiquitin-dependent association with TCOF1 required multiple CK2 motifs, and the most robust signaling was again observed in the presence of all phosphorylation sites (*Figure 6D,E*). Interestingly, NOLC1 variants with five or less CK2 motifs were ubiquitylated by an E3 ligase distinct from KBTBD8 (*Figure 6D*), which did not promote TCOF1-NOLC1 complex formation and has an unknown function (*Figure 6E*).

To determine how many CK2 motifs were required for NOLC1-dependent neural crest specification, we depleted NOLC1 from hESCs using stably expressed shRNAs that targeted the 3′-UTR of its mRNA. Consistent with previous findings (*Werner et al., 2015*), the loss of NOLC1 prevented neural crest specification and induced compensatory expression of forebrain markers, as seen by qRT-PCR, western blotting, or fluorescence microscopy (*Figure 7A–D*). These phenotypes could be rescued by shRNA-resistant wild-type NOLC1, which underscores the specificity of our depletion approach (*Figure 7A–D*). When we introduced NOLC1 variants with increasing numbers of CK2 sites, we found that only those containing seven or more CK2 motifs were able to support neural crest specification (*Figure 7A–D*). As seen with KBTBD8 binding and formation of the NOLC1-TCOF1 platform, wild-type NOLC1 containing all ten CK2 sites was most efficient in driving neural crest specification. Although it is difficult to exclude unknown effects of point mutations, the ability of NOLC1 variants to support neural crest specification therefore closely mirrored their ubiquitylation by CUL3$^{\text{KBTBD8}}$ in cells, and seven or more CK2 motifs need to be present in NOLC1 to allow for modification and function of this ribosome biogenesis factor. We conclude that multisite-

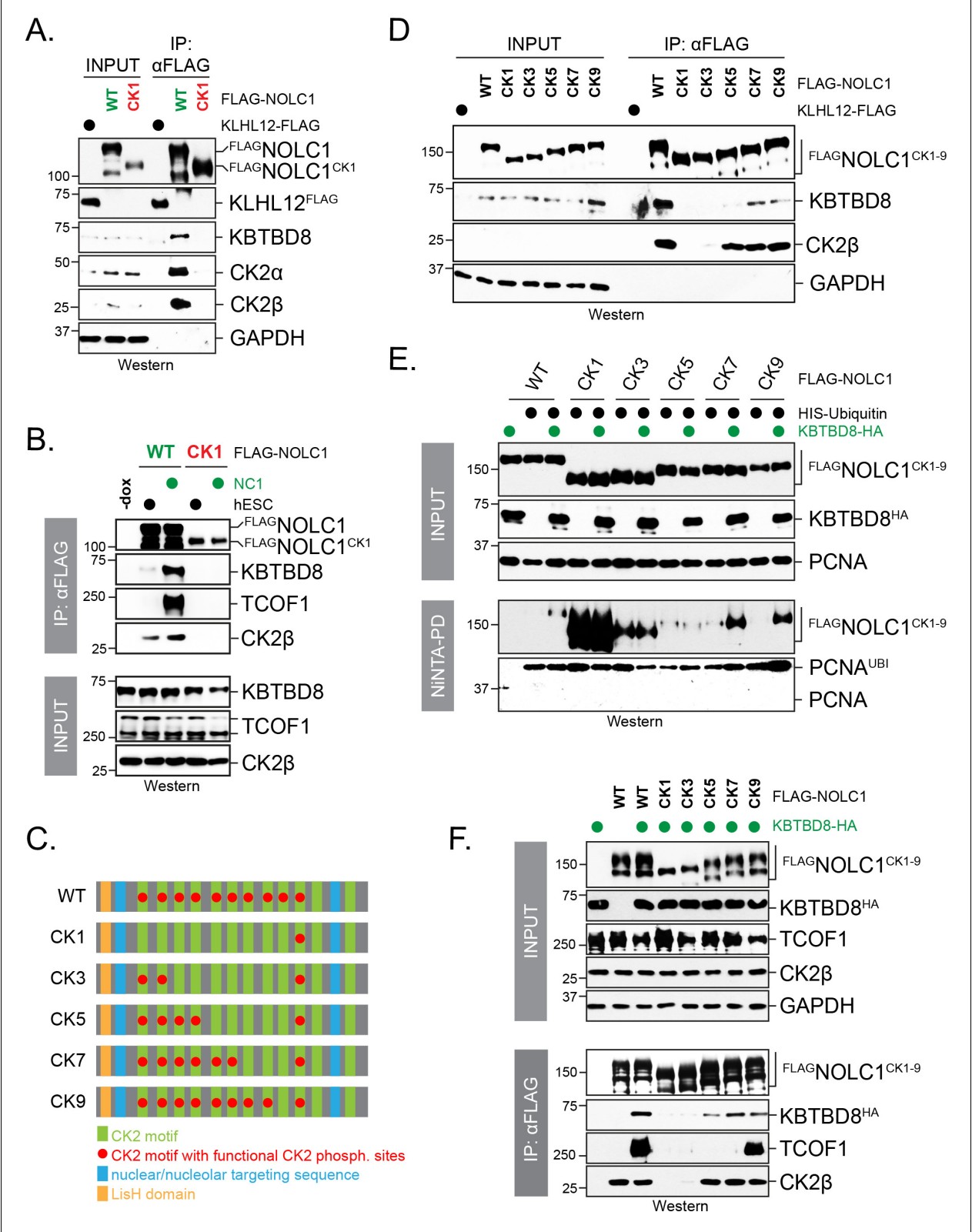

**Figure 6.** Multisite dependency of CUL3[KBTBD8] substrate recognition in cells. (**A**) A single CK2 motif is not sufficient for substrate recognition by KBTBD8 in cells. Wild-type [FLAG]NOLC1 or a mutant containing only a single CK2 motif ([FLAG]NOLC1[CK1]) were affinity-purified from 293 T cells and analyzed for binding to endogenous KBTBD8 and CK2 by western blotting using specific antibodies. (**B**) A single CK2 motif is not sufficient for KBTBD8 and TCOF1 recognition by NOLC1 in differentiating hESCs. [FLAG]NOLC1 was affinity-purified from hESCs or hESCs subjected to 1 day of neural

*Figure 6 continued on next page*

Figure 6 continued

conversion (NC1), and bound endogenous KBTBD8, TCOF1, or CK2 were detected by western blotting using specific antibodies. (**C**) Scheme of CK2-mutants of NOLC1. CK2 motifs that remained functional are labeled with a red dot; the last two CK2 motifs of NOLC1 do not mediate KBTBD8-binding in vitro, as shown above. (**D**) KBTBD8-recognition of NOLC1 depends on multiple CK2 motifs. The indicated NOLC1 variants were affinity-purified from 293 T cells and analyzed for binding to endogenous KBTBD8 or CK2β by western blotting using specific antibodies. Note that wild-type NOLC1, which contains all 10 functional CK2 motifs is the most efficient KBTBD8-recruiter. (**E**) Multisite dependency of NOLC1 ubiquitylation in cells. Ubiquitylated NOLC1 mutants were purified under denaturing conditions from 293 T cells that expressed [HIS]ubiquitin and, where indicated, KBTBD8[HA]. Ubiquitylation of [FLAG]NOLC1 was analyzed by αFLAG-Western; note that NOLC1 mutants containing up to five CK2 motifs are ubiquitylated by an E3 ligase distinct from KBTBD8 (i.e. ubiquitylation is observed in the absence of KBTBD8 expression in these cells). Ubiquitylation of endogenous PCNA was monitored by western blotting using specific antibodies. (**F**) Multisite dependency of TCOF1-NOLC1 complex formation. [FLAG]NOLC1 variants with increasing numbers of CK2 motifs were affinity-purified from 293 T cells that expressed KBTBD8 as indicated. Complex formation with endogenous TCOF1 was monitored by western blotting using specific antibodies. In addition, overexpression of KBTBD8 allowed for KBTBD8 recognition already in the presence of five CK2 motifs, consistent with multisite dependency providing an increase in affinity (i.e. that can in part be overcome by overexpression), rather than creating a specific complex E3 recognition element.

DOI: https://doi.org/10.7554/eLife.35407.011

dependency of CUL3[KBTBD8] governs monoubiquitylation-dependent neural crest specification, a switch-like change in cell fate.

## Discussion

The central role of CUL3 in metazoan development (*McGourty et al., 2016*; *Singer et al., 1999*; *Werner et al., 2015*), and its frequent misregulation in cancer, autism, or hypertension (*Hayward et al., 2017*; *Northcott et al., 2017*; *Rape, 2018*), implies that this E3 ligase needs to be under strict control. Providing a first mechanism of regulation, CUL3[KEAP1] constitutively ubiquitylates the transcription factor NRF2, until this reaction is prevented upon cysteine modification of KEAP1 during oxidative stress (*Furukawa and Xiong, 2005*; *Tong et al., 2006*). Conversely, CUL3[KLHL12] regulates COPII vesicle size only after calcium has been released from the endoplasmic reticulum (*McGourty et al., 2016*). Here, we add another layer of regulation by showing that multisite phosphorylation is required for target recognition by CUL3[KBTBD8], an E3 ligase that controls neural crest specification (*Werner et al., 2015*). While phosphorylation is a common event in promoting substrate recognition by SCF E3 ligases (*Skaar et al., 2013*), this example shows that similar regulation exists for the CUL3 family of E3 ligases.

### Multisite dependency of substrate recognition by CUL3[KBTBD8]

The essential CUL3[KBTBD8]-substrates TCOF1 and NOLC1 each contain 10 or more motifs that can be independently phosphorylated by CK2 and mediate binding to KBTBD8 with comparable efficiency in vitro. Proteomic approaches found most of these CK2 motifs to be phosphorylated in cells (*Bian et al., 2013*), which we confirmed in our own experiments. The CK2 motifs are spread throughout the central disordered domains of TCOF1 and NOLC1, and accordingly, we detected nine distinct monoubiquitylation sites on TCOF1 that were always in proximity to CK2 motifs. While we might have missed cellular ubiquitylation sites, these observations already suggested that multiple CK2 sites can function as CUL3[KBTBD8]-recognition motifs.

Fluorescence polarization measurements showed that KBTBD8 recognizes a single CK2 motif with a dissociation constant of ~1 μM, and steady-state assays indicated that the affinity of KBTBD8 for most CK2 motifs in TCOF1 or NOLC1 is in a similar range. KBTBD8 therefore binds more than 10 distinct motifs in TCOF1 or NOLC1 with an affinity that is akin to that of the E3 ligases SCF[CDC4] or UBR1 to their targets (*Csizmok et al., 2017*; *Tasaki et al., 2009*). However, a single CK2 motif was not sufficient for substrate binding by CUL3[KBTBD8] in cells, where hundreds of proteins compete for access to CK2 (*Bian et al., 2013*). The CUL3[KBTBD8]-dependent binding and monoubiquitylation of NOLC1, formation of TCOF1-NOLC1 complexes, and differentiation of hESCs into neural crest cells required at least seven CK2 motifs. We note that the presence of all 10 CK2 motifs resulted in more robust KBTBD8 recognition and neural crest specification, indicating that most, if not all, E3 binding sites contribute to CUL3[KBTBD8]-substrate ubiquitylation and cell differentiation. We conclude that CUL3[KBTBD8] relies on multisite substrate phosphorylation for optimal target selection.

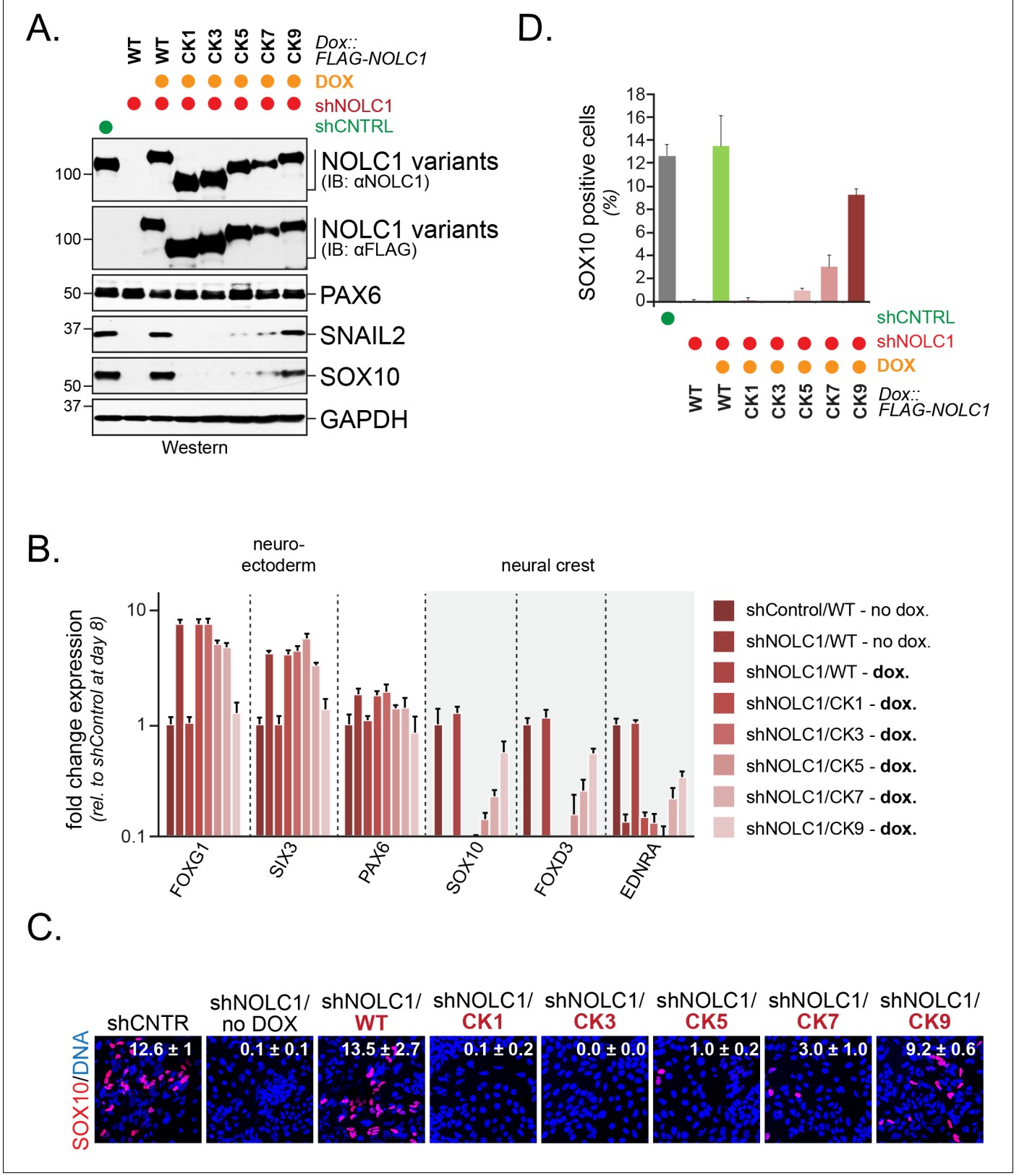

**Figure 7.** Multisite dependency of CUL3$^{KBTBD8}$ during neural crest specification. (A) Multiple CK2 motifs in NOLC1 are required for its role in neural crest specification. Control or NOLC1-depleted H1 hESCs were generated by stable expression of shRNAs. Cells were then reconstituted with doxycycline-inducible wildtype (WT) or CK-mutants of NOLC1 and subjected to neural conversion for 6 days. The outcome of cell differentiation was

*Figure 7 continued on next page*

*Figure 7 continued*

monitored by western blotting using specific antibodies against PAX6 (CNS precursor marker), SNAIL2 and SOX10 (neural crest markers), and GAPDH (control). (B) Multiple CK2 motifs in NOLC1 are required for its role in neural crest specification. H1 hESCs expressing wild type or NOLC1[CK] variants were generated and subjected to neural conversion as described above, and the outcome of differentiation was monitored using qRT-PCR of markers of the CNS (FOXG1, SIX3, PAX6) or the neural crest (SOX10, SNAIL2, FOXD3). (C) Multiple CK2 motifs in NOLC1 are required for its role in neural crest specification, as seen at single-cell resolution by fluorescence microscopy analysis. H1 hESCs expressing wild type or specific NOLC1[CK] mutants were generated and subjected to neural conversion as described above, and neural crest differentiation was monitored by immunofluorescence microscopy using anti-SOX10 antibodies. (D) Quantification of SOX10-positive cells in *Figure 7C* (n = 3, two biological and one technical replicate; error bars denote s.e.m).

DOI: https://doi.org/10.7554/eLife.35407.012

Our results are reminiscent of the yeast CDK inhibitor SIC1, whose recognition by the E3 ligase SCF[CDC4] requires ~5–6 phosphorylation events (*Csizmok et al., 2017*; *Nash et al., 2001*; *Tang et al., 2007*). In a mechanism referred to as allovalency, its multiple E3 binding sites allow the intrinsically disordered SIC1 to rapidly rebind SCF[CDC4] after a single motif dissociated. Similar to SIC1, TCOF1 and NOLC1 have multiple E3 binding motifs that are embedded in an acidic domain of intrinsic disorder (IDRs). TCOF1 and NOLC1, however, differ from SIC1 in that they contain a positively charged carboxy-terminal IDR with nuclear and nucleolar localization signals. Biophysical work suggested that ampholytic IDRs, such as TCOF1 or NOLC1, adopt a compact globular conformation that can undergo rapid structural fluctuations (*Das and Pappu, 2013*). We therefore speculate that most CK2 motifs in TCOF1 and NOLC1 are buried in a collapsed conformation of the IDRs, while some CK2 motifs are frequently, yet stochastically, exposed on the surface of the substrate. The exposed CK2 motifs are recognized by a conserved positively charged loop in KBTBD8 and potentially a second substrate binding site whose existence was implied by the behavior of the KBTBD8[W579A] variant. The compact conformation of its IDRs might limit the propensity of unstructured TCOF1 and NOLC1 to aggregate, while its multiple E3 recognition motifs ensure that ubiquitin-dependent signaling is still possible (*Figure 8*). We propose that multisite dependency of an E3 ligase provides an elegant mechanism to control the activity of intrinsically disordered proteins, which have now been linked to many important cell behaviors (*Shin and Brangwynne, 2017*).

## Implications for ubiquitin-dependent signaling

The mechanism for CUL3[KBTBD8]-substrate recognition implies that multiple residues in TCOF1 or NOLC1 could serve as acceptor sites for monoubiquitylation. As mentioned before, we found that TCOF1 is ubiquitylated in cells on several Lys residues, even though we have not observed multi-monoubiquitylation (i.e. multiple Lys residues in the same substrate are modified with ubiquitin at the same time). It is difficult to rationalize ubiquitylation events that are agnostic to a particular Lys residue with binding of effector proteins that typically recognize both ubiquitin and a proximal substrate motif (*Husnjak and Dikic, 2012*), and indeed, ubiquitin-binding domains have not yet been identified in TCOF1, NOLC1, or their interaction partners. Rather than promoting a specific interaction, we anticipate that the ubiquitin subunits attached to TCOF1 and NOLC1 function as steric block or solubilizing domain to pry open the compact conformation of both substrates. This conformational change could allow TCOF1 and NOLC1 to engage their partners during ribosome biogenesis and neural crest specification. Understanding this potential role of ubiquitylation in cellular signaling will rely on reconstitution of the monoubiquitylation-dependent formation of TCOF1-NOLC1 complexes, which is currently impossible due to difficulties in the purification of these large and unstructured proteins. Why CUL3[KBTBD8] stops after having transferred a single ubiquitin is also an open question that will be an interesting basis for experiments aimed at elucidating mechanisms of monoubiquitylation.

Multisite phosphorylation allows cells to integrate numerous environmental cues into a coherent signaling response, or to introduce ultrasensitivity and threshold effects into signaling circuits (*Gunawardena, 2005*). In case of a distributive kinase, such as CK2, the probability of complete substrate phosphorylation decreases the more sites have already been targeted by the enzyme (*Serber and Ferrell, 2007*). This behavior is likely exacerbated by the large number of CK2 substrates in cells that can act as sponges to titrate the kinase away from TCOF1 and NOLC1. Thus, while the low

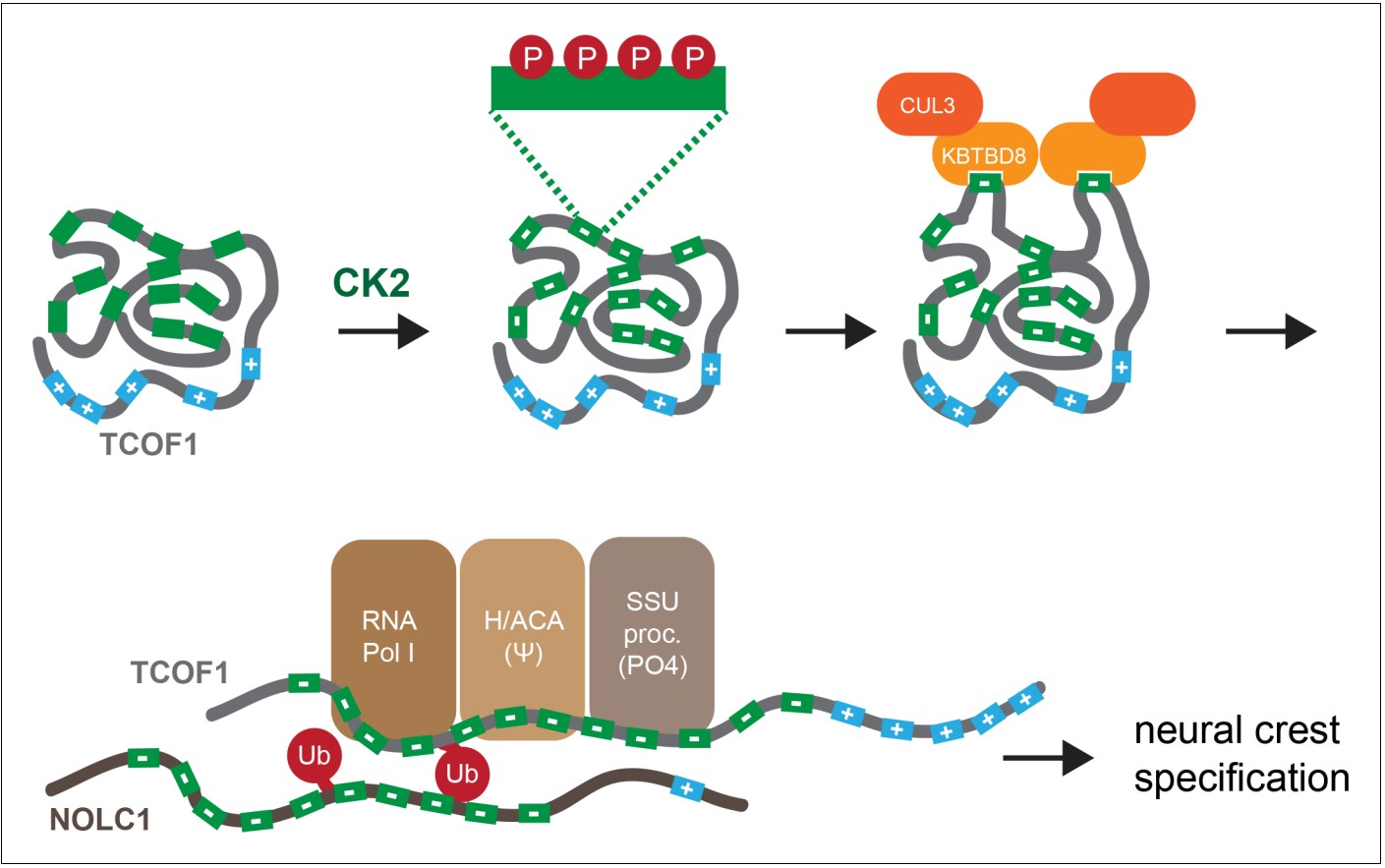

**Figure 8.** Model of multisite-dependent substrate recognition by CUL3[KBTBD8]. CUL3[KBTBD8]-substrates TCOF1 and NOLC1 are first phosphorylated by CK2 in conserved CK2 motifs (12 motifs in TCOF1; 10 motifs in NOLC1), which increases the negative charge of the central intrinsically disordered region (IDR). Phosphorylation likely occurs on multiple Ser residues within a CK2 motif (see insert). While the phosphorylated substrate likely adopts a compact conformation, CK2 motifs are stochastically exposed, allowing for their recognition by KBTBD8. We speculate that the resulting monoubiquitylation of TCOF1 and NOLC1 opens up their conformations to allow for recognition of ribosome biogenesis factors, including RNA polymerase I, the pseudouridylation machinery, and the SSU processome. As shown previously (*Werner et al., 2015*), the subsequent production of new, and likely modified, ribosomes then triggers neural crest specification.

DOI: https://doi.org/10.7554/eLife.35407.013

constitutive activity of CK2 in stem cells might lead to some TCOF1 or NOLC1-phosphorylation, only the increase in CK2 activity observed during nervous system development should enable phosphorylation to the extent required for substrate recognition by CUL3[KBTBD8]. Based on these observations, we propose that multisite dependency of CUL3[KBTBD8] establishes a threshold concentration for CK2, which needs to be surpassed for monoubiquitylation-dependent neural crest specification at specific stages of embryogenesis.

Mutations in *TCOF1* are the major cause of Treacher Collins Syndrome, a craniofacial disease that is characterized by aberrant neural crest specification and survival (*The Treacher Collins Syndrome Colla, 1996*; *Wise et al., 1997*). The mutations in *TCOF1* map throughout the IDR and typically lead to a truncated protein that has fewer phosphorylation sites and hence is less likely to engage in multisite phosphorylation and monoubiquitylation. These mutations also delete carboxy-terminal nuclear and nucleolar localization sequences (*Isaac et al., 2000*), and thus, additionally disrupt targeting of TCOF1 to the nucleolus. Treacher Collins Syndrome is further characterized by an environmental component that strongly affects the phenotypic strength of identical *TCOF1* mutations across patients. It is possible that DNA damage or oxidative stress, which result in TCOF1 modification by kinases other than CK2 (*Ciccia et al., 2014*; *Larsen et al., 2014*; *Sakai et al., 2016*), compete with multisite phosphorylation by CK2. If this were to be the case, mutations that reduce the number of available CK2 sites and environmental conditions that funnel TCOF1 into alternative pathways likely

synergize to prevent monoubiquitylation-dependent neural crest specification. In these cases, our data suggests that inhibition of phosphatases that oppose CK2 or approaches that activate CK2 kinase output could ameliorate phenotypes of *TCOF1* mutation. Conversely, inhibition of CK2 might provide therapeutic benefit against melanoma, a cancer driven by aberrant expression of KBTBD8 (*Hayward et al., 2017*). Our discovery of multisite phosphorylation as key to CUL3$^{KBTBD8}$-substrate recognition and neural crest specification could therefore be an important step towards developing therapies against severe human pathologies.

## Materials and methods

### Plasmids and shRNAs

For transient expression in human cells, the following constructs were generated as previously described (*Werner et al., 2015*): pCMV-3xFlag-TCOF1, pCMV-3xFlag-NOLC1, pCS2-3xHA-TCOF1, pCS2-HA-NOLC1, pCDNA5-KBTBD8(WT, Y74A, F550A, W579A)−3xFlag, pCDNA5-KBTBD8-(WT, Y74A)-HA, pCDNA5-KLHL12-3xFlag, pCS2-6xHis-ubiquitin. KBTBD8$^{5HK}$ (HKH296-298AAA, KK300-301AA) and monomeric KBTBD8$^{L36D/L39D/M42D/N66K}$ were generated by primer extension PCR using nested primers. To generate NOLC1$^{CK1}$ (NOLC1 mutant containing only one functional CK2 phosphorylation motif; all others were inactivated by mutation of two to three amino acids of the CK2 motif to Arg), the central repeat of NOLC1 was synthesized with the desired mutations as gBlock gene fragment (IDT) and cloned with its wild type N- and C-terminus into pCMV-3xFlag by Gibson assembly. To obtain NOLC1$^{CK3}$, NOLC1$^{CK5}$, NOLC1$^{CK7}$, and NOLC1$^{CK9}$, respective DNA fragments were PCR amplified from NOLC1-WT or NOLC1-CK2-1 and assembled by Gibson cloning (*Table 1*).

KBTBD8$^{5HK}$ and monomeric KBTBD8$^{L36D/L39D/M42D/N66K}$ were cloned into pFASTBacHTB with an N-terminal His-tag for expression in SF9 cells using the Bac-to-Bac baculovirus system (Invitrogen) as described for KBTBD8-WT, -F550A, and -W579A (*Werner et al., 2015*). TCOF1 and NOLC1 fragments were cloned with an N-terminal His/FLAG-tag in pET28a (*Table 2*). Plasmids for bacterial expression of CK2α (#27083) and CK2β (#27085) were purchased from Addgene.

For expression in hESCs, KBTBD8$^{5HK}$ was cloned into pENTR1A-EF1α with a C-terminal FLAG-tag and recombined into pLENTI-X1-Hygro as previously described for KBTBD8-WT (*Werner et al.,*

**Table 1.** Mutations introduced in different NOLC1-CK2 mutants.

| NOLC1 CK2 mutant | Amino acids mutated to R |
| --- | --- |
| CK1 | CK2 motif 1: D86, S91<br>CK2 motif 2: S128, S133<br>CK2 motif 3: D168, D172, E176<br>CK2 motif 4: S227, S230<br>CK2 motif 5: S266, S271<br>CK2 motif 6: S322, S326, S332<br>CK2 motif 7: S362, S366, S370<br>CK2 motif 8: S425, S432<br>CK2 motif 9: S470, S47 |
| CK3 | CK2 motif 3: D168, D172, E176<br>CK2 motif 4: S227, S230<br>CK2 motif 5: S266, S271<br>CK2 motif 6: S322, S326, S332<br>CK2 motif 7: S362, S366, S370<br>CK2 motif 8: S425, S432<br>CK2 motif 9: S470, S477 |
| CK5 | CK2 motif 5: S266, S271<br>CK2 motif 6: S322, S326, S332<br>CK2 motif 7: S362, S366, S370<br>CK2 motif 8: S425, S432<br>CK2 motif 9: S470, S477 |
| CK7 | CK2 motif 7: S362, S366, S370<br>CK2 motif 8: S425, S432<br>CK2 motif 9: S470, S477 |
| CK9 | CK2 motif 9: S470, S477 |

DOI: https://doi.org/10.7554/eLife.35407.014

**Table 2.** TCOF1 and NOLC1 fragments cloned with N-terminal His6x-FLAG-tag in pET28a.

| Fragment | Amino acids | Sequence |
|---|---|---|
| TCOF1-F1 | 72–108 | QAKKTRVSDPI STSESSEEEEEAEAE TAKATPRLAST |
| TCOF1-F2 | 189–225 | GMVSAGQ DSSSEDTSSSSDETDVE GKPSVKPAQVKA |
| TCOF1-F3 | 260–298 | AKRAKKPEEESESSEEGSESEEE APAGTRSQVKASEKIL |
| TCOF1-F4 | 329–364 | QTKAGKP EEDSESSSEESSDSEEE TPAAKALLQAKA |
| TCOF1-F5 | 399–431 | AKAQAGKR EEDSQSSSEESDSEEE APAQAKPSG |
| TCOF1-F6 | 466–499 | VQVGKQ EEDSRSSSEESDSDRE ALAAMNAAQVKP |
| TCOF1-F7 | 536–571 | AQVGKW EEDSESSSEESSDSSDGE VPTAVAPAQEKS |
| TCOF1-F8 | 599–638 | VKAEKPMDNSESSEESSDSADSEEAPAAMTAAQAKPALKI |
| TCOF1-F9 | 683–718 | AGGTQRPA EDSSSSEESDSEEE KTGLAVTVGQAKSV |
| TCOF1-F10 | 754–783 | VKAEKQ EDSESSEEESDSEE AAASPAQVKT |
| TCOF1-F11 | 864–893 | P EEDSGSSEEESDSEEEAE TLAQVKPSGKT |
| TCOF1-F12 | 924–954 | AQAGKQ DDSGSSSEESDSDGE APAAVTSAQV |
| NOLC1-F1 | 76–105 | AKKAKKKA SSSDSEDSSEEEEE VQGPPAKK |
| NOLC1-F2 | 118–149 | GKAAAKA SESSSSEESSDDDDEED QKKQPVQK |
| NOLC1-F4 | 207–242 | APKIANGKAA SSSSSSSSSSSSDDSEEE KAAATPKK |
| NOLC1-F5 | 255–283 | VKAATTPTRK SSSSEDSSSDEEEE QKKPM |
| NOLC1-F6 | 313–346 | AVEKQQPV ESSEDSSDESDSSSEEE KKPPTKAVV |
| NOLC1-F7 | 350–384 | TTKPPPAKKAA ESSSDSSDSDSSEDDE APSKPAGT |
| NOLC1-F8 | 415–448 | QKLLTRKA DSSSSEEESSSSEEE KTKKMVATTKP |
| NOLC1-F9 | 460–494 | AKQAPQGSR DSSSDSDSSSSEEEEE KTSKSAVKKK |
| NOLC1-F10 | 510–544 | AKKGKA ESSNSSSSDDSSEEEEE KLKGKGSPRPQA |
| NOLC1-F11 | 552–586 | ALTAQNGKAAKN SEEEEEE KKKAAVVVSKSGSLKK |
| NOLC1-F12 | 619–648 | EKRASSPFRRVR EEEIEVDS RVADNSFDAK |
| NOLC1-F2 WT | 196–212 | DSSSEDTSSSSDETDVE |
| NOLC1-F2 S1A | 196–212 | DASSEDTSSSSDETDVE |
| NOLC1-F2 S3A | 196–212 | DAAAEDTSSSSDETDVE |
| NOLC1-F2 S7A | 196–212 | DAAAEDTAAAADETDVE |
| NOLC1-F2 S1R | 196–212 | DSRSEDTSSSSDETDVE |
| NOLC1-F2 S2R | 196–212 | DSRSEDTSSRSDETDVE |
| TCOF1* | 190–364 | Contains CK2 motif 2–3 |
| NOLC1* | 76–187 | Contains CK2 motif 2–3 |
| TCOF1-R1/2 | 72–225 | Contains CK2 motif 1–2 |
| TCOF1-R1/3 | 72–298 | Contains CK2 motif 1–3 |
| TCOF1-R1/4 | 72–364 | Contains CK2 motif 1–4 |
| TCOF1-R1/6 | 72–499 | Contains CK2 motif 1–6 |
| TCOF1-R1/8 | 72–638 | Contains CK2 motif 1–8 |
| TCOF1-R3/8 | 260–638 | Contains CK2 motif 3–8 |
| TCOF1-R5/8 | 399–638 | Contains CK2 motif 5–8 |
| TCOF1-R7/8 | 536–638 | Contains CK2 motif 7–8 |

DOI: https://doi.org/10.7554/eLife.35407.015

*2015*). NOLC1-WT or CK2 mutant series was cloned with an N-terminal FLAG-tag into pENTR1A and recombined into pINDUCER20 (*Meerbrey et al., 2011*). pLKO1-Puro Mission shRNA targeting KBTBD8 (TRCN0000130280), CKβ (TRCN0000000614), and NOLC1 (TRCN0000293796) were purchased from SIGMA.

## Antibodies

Mouse anti-KBTBD8 antibodies (1 µg/mL in IB) were described previously (*Werner et al., 2015*). Mouse anti-Flag (F1804, clone M2, Sigma, 1:2000 in IB), rabbit anti-PAX6 (Biolegend, PRB-278B, 1:300 in IF), rabbit anti-CK2α (#2656S, Cell Signaling 1:500 in IB) rabbit anti-CK2β (A301-984, Bethyl,1:500 in IB), rabbit anti-CK2 substrate (#8738S, Cell Signaling 1:500 in IB), mouse anti-PCNA clone PC10 (sc56, Santa Cruz, 1:500 in IB), rabbit anti-GAPDH (#2118, clone 14C10, Cell Signaling, 1:10,000 in IB), rabbit anti-ARRB1/2 (#4674, clone D24H9, Cell Signaling, 1:1000 in IB), mouse anti-PKN1 (610687, clone 49/PRK1, BD Bioscience, 1:1000 in IB), rabbit anti-PP1 (#2582, Cell Signaling, 1:100 in IB), rabbit anti-PP2A A (#2041, clone 81G5, Cell Signaling, 1:1000 in IB), rabbit anti-PP2A B (#2290, clone 100C1, Cell Signaling, 1:1000 in IB), rabbit anti-PP2A C (#2259, clone 52F8, Cell Signaling, 1:1000 in IB), rabbit anti-CUL3 (Bethyl, 1:1000 in IB), rabbit anti-NANOG (#3580, Cell Signaling, 1:1000 in IB), goat anti-OCT4 (ac-8628, Santa Cruz, 1:1000 in IB), rabbit anti-SNAIL2 (#9585, clone C19G7, Cell Signaling, 1:500 in IB), rabbit anti-TCOF1 (11003–1-AP, Proteintech, 1:250 in IB), and rabbit anti-NOLC1 (11815–1 P, Proteintech, 1:1000 in IB) antibodies were commercially purchased.

## Proteins

KBTBD8 and KBTBD8 mutant variants (F550A, W579A, 5HK, Δdimer) were purified from SF9 cells 72 hr after transduction. In brief, lysates were prepared in 50 mM sodium phosphate, pH 8, 500 mM NaCl, and 10 mM imidazole by incubation with 200 µg/ml lysozyme and sonication. Lysates were cleared by centrifugation and incubated with Ni-NTA (QIAGEN) for 2 hr at 4°C. Beads were washed with lysis buffer containing 0.1% Triton followed by elution of proteins in 200 mM imidazole in 50 mM sodium phosphate, pH 8, 500 mM NaCl. TEV protease for His-tag removal was added and proteins were dialyzed overnight into 50 mM Tris pH 8.0, 100 mM NaCl, 2 mM DTT, 10% glycerol. Proteins were concentrated and purified by gel filtration using a Superdex 200 column and 20 mM HEPES pH 7.4. 150 mM NaCl, 2 mM DTT. KBTBD8 fractions were concentrated, aliquoted, flash-frozen in liquid nitrogen, and stored at −80°C.

TCOF1 and NOLC1 fragments were purified from E.coli LOBSTR cells. Cell were grown in LB medium to an OD600 of ~0.6 followed by addition of 0.5 mM IPTG to induce protein production at 37°C for 3 hr. Cells were harvested and lysed in 15 mL of lysis buffer A (50 mM TRIS pH 8.0, 50 mM NaCl, 1 mM PMSF, 1 mM EDTA, 3 U/mL Benzonase, 5 mg/mL Lysozyme) per 1.5 L of culture. After incubation for 30 min at 4°C, 10 mL of lysis buffer B (75 mM TRIS pH 8.0, 1425 mM NaCl, 1.5 mM PMSF, 15 mM β-mercaptoethanol, 30 mM imidazole per 1.5 L culture were added followed by sonication on ice. Lysates were cleared by centrifugation. His-tagged TCOF1 and NOLC1 fragments were isolated using NI-NTA resin (Qiagen), washed with wash buffer (50 mM TRIS pH 8.0, 500 mM NaCl, 1 mM PMSF, 5 mM β-mercaptoethanol, 20 mM imidazole), eluted in elution buffer (50 mM TRIS pH 8.0, 300 mM NaCl, 1 mM PMSF, 5 mM β-mercaptoethanol, 250 mM imidazole), and transferred into storage buffer (50 mM TRIS pH 8.0, 300 mM NaCl, 1 mM PMSF, 5 mM β-mercaptoethanol, 15% glycerol) by three cycles of concentration/dilution using Amicon Ultra centrifugal filter units (EMD Millipore, MWCO 3 kDa). Longer TCOF1 fragments (TCOF1-R1/6 and TCOF1-R1/8) were further purified by molecular sieving over a Superdex 200 column in 20 mM HEPES, pH 7.4, 150 mM NaCl, 1 mM DTT, 10% glycerol, followed by concentration of protein-containing fractions. Concentrated proteins were aliquoted, flash-frozen in liquid nitrogen, and stored at −80°C.

Nedd8-modified CUL3/RBX1, CK2α, CK2α/β, and E1/UBE2D3 were purified as previously described (*Jin et al., 2012*; *Meyer and Rape, 2014*; *Turowec et al., 2010*). Initial CUL3/RBX1 ~ Nedd8 purifications were a kind gift from Brenda Schulman.

## Mammalian cell culture and transfection

Human embryonic kidney (HEK) 293 T cells were maintained in DMEM with 10% fetal bovine serum. Plasmid transfections of HEK 293 T cells were with PEI or TransIT-293 Reagent (MIR207, Mirus) according to the manufacturer's instructions. For in vivo ubiquitylation assays, a plasmid ratio of 7.5: 2.5: five for His-ubiquitin: KBTBD8: TCOF1/NOLC1 was used. 293 T cells were routinely tested for mycoplasma infections.

## hESC cell culture

Human embryonic stem (hES) H1 cells were obtained from the Wisconsin stem cell bank and maintained under feeder-free conditions on Matrigel-coated plates (#354277, BD Biosiences) in mTeSR 1, (#05871/05852, StemCell Technologies Inc.). hES H1 cells were routinely passaged with collagenase (#07909, StemCell Technologies Inc.). hESCs were routinely tested for mycoplasma infection.

## Lentiviral production and infection

Lentiviruses were produced in 293 T cells by co-transfection of lentiviral constructs with packaging plasmids (Addgene) for 48 hr. Viruses were collected and filtered through a 0.45 µm filter. For pIND-CUER20 and pLENTI constructs, viruses were concentrated using LENTI-X concentrator (Takara) and directly used for infection. For pLKO1 constructs, supernatant was aliquoted and stored at −80°C. HEK293T cells were transduced in polybrene (6 µg/ml) and selected with 1 µg/mL puromycin for 5 days. hES H1 cells were transduced as single-cell suspensions (prepared by treatment with accutase; $2.5 \times 10e5$ cells per well of a six-well plate) and in the presence of polybrene (6 µg/ml) and 10 µM Y-27632 ROCK inhibitor for 90 min. In case of pINDUCER20 and pLENTI constructs, cells were centrifuged during infection period at 30°C and 1000 g. After infection, medium was changed to mTESR containing 10 µM Y-27632 ROCK inhibitor. Selection of cells hES H1 cells was performed for 7 days using appropriate antibiotics (pLKO1: 1 µg/mL Puromycin, pLENTI-Hygro: 200 µg/mL hygromycin, pINDUCER20: 200 µg/mL G418).

## Neural conversion of hESCs

Neural induction of hES H1 cells expressing different shRNA or protein constructs was performed using STEMdiff^TM Neural Induction Medium (#05831, StemCell Technologies Inc.) in combination with a monolayer culture method according to the manufacturer's technical bulletin (#28044) and as previously described (*Werner et al., 2015*). In brief, single cell suspensions were prepared by treatment of hES cells with accutase ($1.25 \times 10^6$ cells were seeded per well of a six-well plate in STEMdiff Neural Induction Medium supplemented with 10 µM Y-27632 ROCK inhibitor. Neural induction was performed for indicated time periods with daily medium change. To determine the effect of CK2 or PP1/PP2A inactivation on neural conversion, indicated concentrations of CX4945 or okadaic acid were added throughout the differentiation experiment. For NOLC1 rescue experiments, to induce expression of NOLC1 variants, 1 µg/mL doxycycline was added to indicated conditions for 12 hr before the start and until the end of the differentiation experiment.

## Immunoprecipitation

Small scale anti-FLAG and anti-HA immunoprecipitations (IPs) were performed from extracts of HEK293T cells transiently expressing indicated constructs. Lysis was in two pellet volumes of 20 mM HEPES pH 7.3, 50 mM NaCl 110 mM KOAc, 2 mM Mg(OAc)$_2$, 5 mM EDTA, 5 mM EGTA, 0.2% NP-40, and protease inhibitors (Roche) on ice. Lysates were sonicated, cleared by centrifugation, passaged through a 0.45 µm membrane filter, and incubated with ANTI-FLAG-M2 agarose (Sigma) or EZview Red Anti-HA affinity gel for 2 hr at 4°C. After washing with lysis buffer, FLAG-tagged or HA-tagged protein complexes were eluted with sample buffer and analyzed by SDS page and immunoblotting using indicated antibodies. For immunoprecipitation of ^FLAGNOLC1 from self-renewing and differentiating hESCs (hESCs undergoing neural conversion for 1d), 1 × 15 cm dishes of cells induced for flag-NOLC1 expression for 72 hr were used as starting material.

## Mass spectrometry and compPASS analysis

For mass spectrometry analysis, ^FLAGNOLC1, and KBTBD8-5HK^FLAG immunoprecipitates (originating from 20 × 15 cm of transfected 293 T cells) were prepared and analyzed as previously described (*Werner et al., 2015*). Samples were analyzed by the Vincent J. Coates Proteomics/Mass Spectrometry Laboratory at UC Berkeley and compared with ~150 reference immunoprecipitations against different FLAG-tagged bait proteins using a Java script programmed according to the CompPASS software suite (*Huttlin et al., 2015*). To identify regulators of CUL3^KBTBD8 function, we compared interactors found in IPs of its key targets ^FLAGTCOF1 (published in [*Werner et al., 2015*]) and ^FLAG-NOLC1 (this study). For both ^FLAGTCOF1 and ^FLAGNOLC1, three independent IPs were compared as replicates against the reference IPs. Thresholds for high confidence interaction partners (HCIPs)

were top 5% of interactors with highest Z-score and highest WD score. We then plotted relative TSCs of HCIPs (normalized to 1000 TSC of bait) found in both $^{FLAG}$TCOF1 and $^{FLAG}$NOLC1 IPs. To assess the impact of the mutation of the 5HK loop on the KBTBD8 interaction network, we analyzed the abundance of KBTBD8 interactors (TSCs normalized to 1000 TSC of bait) found in IPs of different KBTBD8 variants by cluster analysis. For this we used data sets from KBTBD8-WT$^{FLAG}$, KBTBD8-Y74A$^{FLAG}$, KBTBD8-F550A$^{FLAG}$, and KBTBD8-W579A$^{FLAG}$ IPs (*Werner et al., 2015*) and from KBTBD8-5HK$^{FLAG}$ (this study).

For determination of TCOF1 ubiquitylation sites, His-ubiquitin, $^{HA}$TCOF1, and KBTBD8$^{FLAG}$ were co-expressed in HEK 293 T cells and ubiquitylated TCOF1 conjugates were isolated by NiNTA pull down. Fractions were separated by SDS-Page, stained with Coomassie, and gel pieces containing TCOF1 ~Ub conjugates were excised (appropriate gel pieces were identified by simultaneous immunoblotting of a small NiNTA fraction run on a different gel). Proteins were subjected to in-gel digestion using trypsin or gluC and analyzed by the Vincent J. Coates Proteomics/Mass Spectrometry Laboratory at UC Berkeley. Protein identification was done with IntegratedProteomics Pipeline (IP2, Integrated Proteomics Applications, Inc. San Diego, CA) using a false positive rate at the peptide level of 0.25%. Individual spectra for ubiquitylated peptides were inspected to ensure that fragments defined the ubiquitylation site.

## Modeling of the KBTBD8 structure

A predicted 3D model of the KBTBD8 structure was obtained from the RaptorX web server. The derived model was processed and analysed using PyMOL (The PyMOL Molecular Graphics System, Version v1.3r1 Schrödinger, LLC.). The conservation analysis was done using the ConSurf server (*Landau et al., 2005*).

## In vitro phosphorylation assays

To ensure equal concentrations of purified TCOF1 and NOLC1 fragments for in vitro assays, protein concentrations were determined using the Pierce BCA Protein Assay Kit (#23225, Thermo Scientific). TCOF1 and NOLC1 fragments in concentrations ranging from 3.5 to 500 μM were phosphorylated for 90 min at 30°C using CK2 (#P6010, NEB, or self-purified as described above; $c_{final}$ = 0.1-2 μM) and ATP (0.4 mM-2 mM) in PK buffer (#B6022S, NEB). For unphosphorylated controls, either ATP or CK2 was substituted with $H_2O$. Phosphorylation reactions were quenched by addition of EDTA to a final concentration of 10 mM and subsequently used for binding assays.

## In vitro binding assays

For KBTBD8 binding assays, ANTI-FLAG-M2 agarose (Sigma) beads were equilibrated in binding buffer (20 mM HEPES pH 7.3, 110 mM KOAc, 2 mM Mg(OAc)$_2$, 5 mM EGTA, 0.05% NP-40). Flag-tagged TCOF1 and NOLC1 fragments from phosphorylation reactions were added into a 200 μL reaction and incubated rotating for 1 hr at RT. Unbound protein fragments were rinsed three times with 400 μL binding buffer. KBTBD8 and KBTBD8 mutant protein (5-15 μg/sample) were added to the beads and incubated for 1 hr at RT on a rotator. After washing with binding buffer, bound protein was eluted with sample buffer and analyzed by SDS page and immunoblotting using indicated antibodies.

For TCOF1 fragment binding assays, Protein G agarose (Sigma) beads were equilibrated in binding buffer and incubated with KBTBD8 or KBTBD8 mutant protein and mouse anti-KBTBD8 antibody for 1 hr at RT. After washing out unbound protein with binding buffer, phosphorylated or unphosphorylated protein fragments were added to the reaction and incubated for 1 hr on a rotator. After washing out unbound fragments with binding buffer, bound complexes were eluted with sample buffer and analyzed by SDS page and immunoblotting using indicated antibodies.

## Determination of the ultrasensitivity of the KBTBD8-TCOF1 interaction

Reactions of 4 μM TCOF1 fragment R1/8 in the presence or absence of tenfold excess of competitor peptide (TCOF1-R1/2) were phosphorylated with 1 mM ATP using indicated CK2 concentrations in PK buffer (#B6022S, NEB) at 30°C for 1 hr. Phosphorylation reactions were stopped via addition of EDTA to a final concentration of 10 mM.

Protein G agarose (Sigma) beads were equilibrated in binding buffer and incubated with KBTBD8 (5 μg/sample) and mouse anti-KBTBD8 antibody for 1 hr at RT. After washing out unbound protein, the phosphorylation reactions were added to immobilized KBTBD8 to a final concentration of 450 nM and incubated at RT for 1 hr. The binding reactions were washed, eluted in sample buffer and analyzed by SDS page and immunoblotting using indicated antibodies. ImageJ was used for signal quantification of two individual experiments. GraphPad was used for subsequent data processing and analysis using the specific binding fit with Hill slope equation and data was plotted with the standard error of the mean.

## Fluorescence polarization assays

TAMRA-labeled TCOF1 peptide (5,6-[TAMRA]AGQDSSSEDTSSSSDETDVEGKP, bioSyntesis) and competitor peptides were phosphorylated as described above. $K_D$ measurements were done at 100 nM TAMRA-TCOF1 peptide and indicated concentrations of KBTBD8 and KBTBD8 mutant proteins in triplicates and duplicates, respectively. Competition assays were done in duplicates with 5 μM KBTBD8 and 2.5 μM KBTBD8 Δdimer and indicated concentrations of TCOF1 competitors. Fluorescence polarization signal was measured using a BioTek Synergy H4 plate reader after 30 min incubation at RT in the dark. Data was processed and analyzed in GraphPad using the specific binding fit with Hill slope equation and plotted with the standard error of the mean.

## In vitro ubiquitylation assays

For in vitro ubiquitylation assays, 2 μM of His6x-FLAG-TCOF1-R1/2 and His6x-FLAG-TCOF1-R1/8 proteins were phosphorylated as described above and bound to anti-FLAG agarose in a 100 μL reaction volume at RT for 1 hr. Beads were washed with binding buffer (20 mM HEPES pH 7.3, 50 mM NaCl, 110 mM KOAc, 2 mM Mg(OAc)$_2$, 1 mM EGTA, 0.1% NP-40) and incubated with 2 μM of KBTBD8 in a 100 μL reaction volume at RT for 1 hr, followed by two washes of binding buffer. Liquid was removed completely from beads and 20 μL ubiquitylation mix (0.5 μM Ube1, 1.0 μM UBE2D3, 40.0 μM ubiquitin, indicated concentrations of Cul3/Rbx1 ~ Nedd8, and 1 mM ATP in 20 mM TRIS pH 7.5, 50 mM NaCl, 10 mM MgCl$_2$) were added. Reactions were incubated at 30°C for 1 hr and then quenched by addition of 20 μL of 2X sample buffer. Eluates were analyzed by SDS-PAGE followed by immunoblotting. Of note, immobilizing the substrates on FLAG-agarose and pre-binding KBTBD8 was required for specific substrate modification.

## In vivo ubiquitylation assays

For detection of TCOF1 and NOLC1 ubiquitylation, HEK293T cells were transfected with [6xHis]Ubiquitin and KBTBD8[FLAG]. To determine the effect of CK2 inactivation, cells were stably transfected with control or shRNAs targeting CK2β before plasmid transfection and treated with 20 μM CX4945 24 hr after plasmid transfection. To determine the effect on inactivation of PP1 and PP2A phosphatases, cells were treated with 1 μM okadaic acid for 30 min prior to lysis. Cells were harvested 48 hr after transfection, washed with PBS, lysed in 8M urea, 50 mM Tris-HCl pH 8.0, 10 mM imidazole and sonicated. [6xHis]Ubiquitin conjugates were purified by incubation and rotation with NiNTA agarose for 1 hr at room temperature. Beads were washed 2x with lysis buffer and 2x with lysis buffer at pH 6.3. Ubiquitin conjugated were eluted in sample buffer and ubiquitylated proteins were detected by SDS page followed by immunoblotting.

## Quantitative real time PCR analysis

For qRT-PCR analysis, total RNA was extracted and purified from cells using the RNeasy Mini Kit (Qiagen, catalogue no. 74104) and transcribed into cDNA using the RevertAid first strand cDNA synthesis kit (#K1621, Thermo Scientific). Gene expression was quantified by Maxima SYBR Green/Rox qPCR (#K0221, Thermo scientific) on a StepOnePlus Real-Time PCR System (Applied Biosystems). Nonspecific signals caused by primer dimers were excluded by dissociation curve analysis and use of non-template controls. To normalize for loaded cDNA, RPS6 was used as endogenous control. Gene-specific primers for qRT–PCR were ordered pre-designed from Integrated DNA Technologies. Primer sequences are summarized in *Table 3*.

**Table 3.** qPCR primers used in this study

| qPCR primer | Sequence |
| --- | --- |
| RPS6 right | TGA TGT CCG CCA GTA TGT TG |
| RPS6 left | TCT TGG TAC GCT GCT TCT TC |
| TFAP2 right | ATT GAC CTA CAG TGC CCA GC |
| TFAP2 left | ATG CTT TGG AAA TTG ACG GA |
| PAX6 right | CAC ATG AAC AGT CAG CCA ATG |
| PAX6 left | GGC CAG TAT TGA GAC ATA TCA GG |
| EDNRA right | CAT GAC TTG TGA GAT GTT GAA CAG |
| EDNRA left | CTG TTT TTG CCA CTT CTC GAC |
| FOXD3 right | TTG ACG AAG CAG TCG TTG AG |
| FOXD3 left | TCT GCG AGT TCA TCA GCA AC |
| SIX3 right | CGA GGG GAG TGG ACA CTT |
| SIX3 left | ATG CCG CTC GGT CCA AT |
| SNAI2 right | TGA CCT GTC TGC AAA TGC TC |
| SNAI2 left | CAG ACC CTG GTT GCT TCA A |
| SOX10 right | CTT TCT TGT GCT GCA TAC GG |
| SOX10 left | AGC TCA GCA AGA CGC TGG |
| FOXG1 right | TGA ATG AAA TGG CAA AGC AG |
| FOXG1 left | TGC AAT GTG GGG AGA ATA CA |

DOI: https://doi.org/10.7554/eLife.35407.016

## Immunofluorescence microscopy

For immunofluorescence analysis, hES H1 cells or hES H1 cells expressing different shRNA and/or proteins and subjected to neural conversion for indicated time periods were seeded on Matrigel-coated converslips using accutase, fixed with 3.7% formaldehyde for 10 min, permabilized with 0.1% Triton for 20 min, and stained with anti-SOX10 antibodies and Hoechst. Random Images were taken using a Zeiss LSM 710 confocal microscope with a 20x objective and processed using ImageJ. SOX10-positive cells and total nuclei were counted (~150–200 cells per condition for each of the three replicates) and average percentage of SOX10-positive cells was determined.

## Acknowledgements

We thank Julia Schaletzky for comments on the manuscript, and Brenda Schulman for the generous gift of Nedd8-modified CUL3. We are grateful to all members of our laboratory for continuing discussions and suggestions. AW is a recipient of a NIDCR/NIH pathway to independence award (K99DE025314). MR is the K Peter Hirth Chair of Cancer Biology at UC Berkeley and an Investigator of the Howard Hughes Medical Institute.

## Additional information

### Competing interests

Michael Rape: Reviewing Editor, eLife. is founder of and consultant to Nurix, a biotechnology company acting in the ubiquitin space. The work at Nurix does not overlap with the current study. The other authors declare that no competing interests exist.

## Funding

| Funder | Grant reference number | Author |
|---|---|---|
| Howard Hughes Medical Institute | | Michael Rape<br>Regina Baur |
| National Institutes of Health | K99DE025314 | Achim Werner |

The funders had no role in study design, data collection and interpretation, or the decision to submit the work for publication.

## Author contributions

Achim Werner, Conceptualization, Data curation, Formal analysis, Funding acquisition, Writing—review and editing; Regina Baur, Conceptualization, Data curation, Formal analysis, Writing—review and editing; Nia Teerikorpi, Deniz U Kaya, Data curation, Formal analysis; Michael Rape, Conceptualization, Resources, Formal analysis, Supervision, Funding acquisition, Investigation, Writing—original draft, Project administration, Writing—review and editing

## Author ORCIDs

Regina Baur (ID) https://orcid.org/0000-0001-5104-4888
Deniz U Kaya (ID) https://orcid.org/0000-0003-4767-8655
Michael Rape (ID) https://orcid.org/0000-0003-4849-6343

## Decision letter and Author response

Decision letter https://doi.org/10.7554/eLife.35407.019
Author response https://doi.org/10.7554/eLife.35407.020

# Additional files

## Supplementary files

• Transparent reporting form
DOI: https://doi.org/10.7554/eLife.35407.017

## Data availability

All data generated or analysed during this study are included in the manuscript and supporting files.

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
