## [Decision Letter]

Thank you for submitting your article "Multisite dependency of an E3 ligase controls monoubiquitylation-dependent cell fate decisions" for consideration by *eLife*. Your article has been reviewed by Ivan Dikic as the Senior Editor, a Reviewing Editor, and two reviewers. The following individuals involved in review of your submission have agreed to reveal their identity: Simona Polo (Reviewer #1).

The reviewers have discussed the reviews with one another and the Reviewing Editor has drafted this decision to help you prepare a revised submission.

Summary:

E3 ligases serve prominent roles in the development of multicellular organisms and this necessitates that their activity be tightly controlled both spatially and temporally. E3 ligase-substrate interactions often depend of substrate phosphorylation and therefore the corresponding kinase activity can provide a timing signal for substrate ubiquitylation and the processes they regulate. In the submitted manuscript, Werner et al. demonstrate that the activity of the CK2 kinase, which increases during embryogenesis, turns on the recognition and the monoubiquitylation of NOLC1 and TCOF1 by the E3 ligase CUL3^KBTBD8^. These two substrate proteins are intimately involved in neural crest differentiation once mono-ubiquitinylated. Multisite phosphorylation is carried out by the CK2 kinase, which recognizes and phosphorylates several CK2 consensus sites in both TCOF1 and NOLC1. Using in vitro approaches to evaluate binding affinities, the authors nicely show the relevance of the CK2 mediated phosphorylation for KBTBD8-TCOF1 and KBTBD8-NOLC1 interaction. They further demonstrate that a single phosphorylation event is enough to positively influence the interaction in vitro while more than 7 phosphorylation events are necessary to allow the interaction in vivo, and the consequent ubiquitylation. Finally, the authors show that multiple CK2 motifs are required for optimal recognition by CUL3^KBTBD8^ and for neural crest specification. They hypothesize that multisite dependency allows cells to convert a gradual increase in kinase expression (embryonic CK2) into a sharp activation of the signaling output (ubiquitylation of the TCOF1-NOLC complex formation).

Overall the study provides compelling evidence in support of the importance of multi-site phospho-recognition of TCOF1/NLCO1 by KBTBD8 and its ability to impart switch-like/ultrasensitive control of neural crest differentiation under the control of CK2 activity. In addition, the experimental results are fully consistent with the proposed binding model. However, additional experiments would strengthen the manuscript by providing additional support to their claims and to rule out alternate models.

Essential revisions:

1) The authors mention that CK2 expression changes during embryogenesis (Subsection “CK2 kinase is required for CUL3^KBTBD8^-dependent neural crest specification”), but never determine the expression profile of CK2 in their assay, the neural crest specification system from hESCs. This should be performed in a time course experiment, together with the expression profile of PP1 and PP2A, the phosphatases that in 293T cells oppose CK2 and promote substrate binding and monoubiquitylation. Similarly, the authors should test the importance of phosphatase activity in neural crest specification, using siRNA or inhibitors as they did for CK2 (Figure 2).

2) The effect of multisite phosphorylation on monoubiquitylation is evaluated only in 293T cells. It would be important to determine the phosphorylated and ubiquitylated state of NOLC1 WT and NOLC1^CK1^ to NOLC1^CK9^ in the hESCs model, to be able to attribute the defect observed in neural crest specification to the phosphorylation threshold and subsequent monoubiquitylation.

3) Homology modeling and conservation analysis Figure 3A. What is the diversity of species used in the analysis? My concern is that everything appears conserved if too closely related species are compared. If the KBTBD8 protein is only present in higher organisms, then conservation analysis is not that informative. The fact that the authors found mutations that work reduces my concern, but it would be easy to beef this analysis up. In addition, the authors should include the homology model coordinates as supplementary information. A more informative analysis would be to project conservation onto the surface of their homology model. Does this identify a hot spot of conservation suggestive of a degron biding site?

4) Figure 5F. All the deletion mutants are generated by deletion from one end of the protein. One possible explanation for the behavior of the system (that needs to be ruled out) is that one extreme end of TCOF1 houses a secondary binding site for KBTBD8, which is responsible for the apparently all or none binding behavior. This could be ruled out by generating and testing a limited number of deletion mutants of TCOF1 starting from the other terminus.

---

## [Author Response]

Essential revisions:1) The authors mention that CK2 expression changes during embryogenesis (Subsection “CK2 kinase is required for CUL3^KBTBD8^-dependent neural crest specification”), but never determine the expression profile of CK2 in their assay, the neural crest specification system from hESCs. This should be performed in a time course experiment, together with the expression profile of PP1 and PP2A, the phosphatases that in 293T cells oppose CK2 and promote substrate binding and monoubiquitylation. Similarly, the authors should test the importance of phosphatase activity in neural crest specification, using siRNA or inhibitors as they did for CK2 (Figure 2).

We have followed the advice of the reviewers and determined the expression profiles of CK2, PP1, and PP2A in differentiating hESCs using RNAseq and Western blotting (new Figure 1—figure supplement 1A, B). While mRNA levels of CK2, PP1, and PP2A subunits do not change dramatically during neural conversion, we did observe that the protein levels of PP1A and PP2A C decreased over the time course of differentiation. Western blotting showed a corresponding slight, but reproducible, increase in CK2 substrate phosphorylation at the first day of differentiation, i.e. at the time when neural crest specification is initiated. Thus, similar to the situation in mouse embryos, where CK2 levels peak at day 11/12 of development, our data suggests that in in vitro differentiation systems CK2 output increases at the time of neural crest specification.

We also monitored the efficiency of neural crest formation upon PP1 and PP2A inhibition. Given the essential cell cycle functions of both phosphatases (Wang et al., 2015), siRNA experiments unfortunately are impossible to conduct. As an alternative strategy, we treated cells with low concentrations of okadaic acid, a small molecule that inhibits both PP1 and PP2A. Although at high concentrations of okadaic acid face the same problems as siRNA-mediated co-depletion of PP1 and PP2A (i.e. hESCs die at concentrations above 1nM), lower levels of this PP1/PP2A inhibitor allowed hESCs to survive; more importantly, under these conditions, we observed a dose-dependent stimulation of neural crest specification (new Figure 2—figure supplement 1).

2) The effect of multisite phosphorylation on monoubiquitylation is evaluated only in 293T cells. It would be important to determine the phosphorylated and ubiquitylated state of NOLC1 WT and NOLC1^CK1^ to NOLC1^CK9^ in the hESCs model, to be able to attribute the defect observed in neural crest specification to the phosphorylation threshold and subsequent monoubiquitylation.

We shared the reviewers’ wish to interrogate the phosphorylation and ubiquitylation status of the NOLC1 mutants in differentiating hESCs. However, such experiments are cost-prohibitive (we calculated that due to the high cost of hESC and neural conversion medium, the suggested ubiquitylation analysis would cost ~20,000$). Our previous work had shown that 293T cells can be used to study stem cell-specific E3 ligases and their regulation (Jin et al., 2012; McGourty et al., 2016; Werner et al., 2015); moreover, the ability of NOLC1^CK1^ to NOLC1^CK9^ variants to drive neural crest specification was completely consistent with binding and ubiquitylation studies of wildtype NOLC1 performed in 293T cells (Werner et al., 2015). Thus, we hope that the reviewers agree that the insight gained by repeating ubiquitylation studies in hESCs does not justify the large financial investment.

Our previous work had shown that NOLC1-binding to CUL3^KBTBD8^ strictly correlates with ubiquitylation of NOLC1 and its ability to induce neural crest specification (Werner et al., 2015). To provide further evidence that CUL3^KBTBD8^ shows multisite dependency in hESCs, we therefore asked whether more than one CK2 site is required for recognition of NOLC1 by CUL3^KBTBD8^ in hESCs. We expressed low levels of wildtype NOLC1 with 10 CK2 motifs or a mutant that contains a single CK2 site (NOLC1^CK1^) in self-renewing hESCs, where CUL3^KBTBD8^ is not required, or in differentiating hESCs, where CUL3^KBTBD8^ is essential to monoubiquitylate NOLC1 and drive neural crest specification. Consistent with the phosphorylation-dependent regulation of CUL3^KBTBD8^-substrate recognition, NOLC1 engaged endogenous KBTBD8 and TCOF1 only upon initiation of differentiation. By contrast, NOLC1^CK1^ did not show any of these interactions (new Figure 6B). This confirms that substrate engagement by CUL3^KBTBD8^ and downstream ubiquitin-dependent signaling (i.e. TCOF1 binding) require multiple CK2 motifs also in differentiating hESCs.

3) Homology modeling and conservation analysis Figure 3A. What is the diversity of species used in the analysis? My concern is that everything appears conserved if too closely related species are compared. If the KBTBD8 protein is only present in higher organisms, then conservation analysis is not that informative. The fact that the authors found mutations that work reduces my concern, but it would be easy to beef this analysis up. In addition, the authors should include the homology model coordinates as supplementary information. A more informative analysis would be to project conservation onto the surface of their homology model. Does this identify a hot spot of conservation suggestive of a degron biding site?

We followed the advice of the reviewers and improved both our homology and modeling analysis. We took KBTBD8 variants from 54 different vertebrate species as the basis for a new sequence comparison (Figure 3—figure supplement 1), identified highly conserved residues, and mapped these onto a structural model of the Kelch repeats on KBTBD8. This analysis showed that the 5HK loop is very highly conserved – in fact, with the exception of a single His-residue, this loop is invariant (new Figure 3A). As mentioned by the reviewers, this observation is highly suggestive of a degron binding site, which in fact might include an additional conserved positively charged surface in close proximity to the 5HK loop.

4) Figure 5F. All the deletion mutants are generated by deletion from one end of the protein. One possible explanation for the behavior of the system (that needs to be ruled out) is that one extreme end of TCOF1 houses a secondary binding site for KBTBD8, which is responsible for the apparently all or none binding behavior. This could be ruled out by generating and testing a limited number of deletion mutants of TCOF1 starting from the other terminus.

To address this concern, we generated TCOF1 fragments that lacked amino-terminal CK2 motifs, and instead contained more carboxy-terminal CK2 motifs. We found that fragments with two or four CK2 motifs behaved identical in KBTBD8 binding irrespective of whether the fragments were obtained from the amino- or carboxy-terminus of TCOF1 (new Figure 5—figure supplement 1). We also noted a very interesting behavior of another C-terminal fragment, which contained an inactive CK2 motif embedded between two and three active CK2 motifs, respectively (R3/8; the sixth CK2 motif contains two Arg residues and is therefore unable to bind CK2, see Figure 4D). This fragment bound KBTBD8 with an affinity comparable to R1/3, an amino-terminal fragment with three active CK2 motifs. The R3/8 fragment therefore provided the insight that it is multisite phosphorylation in consecutive CK2 motifs that is required for recognition by CUL3^KBTBD8^. We have added this data to our manuscript, and we thank the reviewers for motivating us to conduct this interesting experiment.